



# OH reactivity from different tree species: Investigating the missing reactivity in a boreal forest

Arnaud P. Praplan[1], Toni Tykkä[1], Simon Schallhart[1], Virpi Tarvainen[1], Jaana Bäck[2], and Heidi Hellén[1]

[1]Atmospheric Composition Research, Finnish Meteorological Institute, P.O. Box 503, 00101 Helsinki, Finland
[2]Institute for Atmospheric and Earth System Research/Forest Sciences, Faculty of Agriculture and Forestry, P.O. Box 27, 00014 University of Helsinki, Finland

**Correspondence:** A. P. Praplan (arnaud.praplan@fmi.fi)

**Abstract.**

In forested area, a large fraction of total hydroxyl radical (OH) reactivity remain unaccounted for. Very few studies have been looking at total OH reactivity from biogenic emissions and its variations. In the present study, we investigate the total OH reactivity from three common boreal tree species (Scots pine, Norway spruce, and Downy birch), by comparing it with

5 the calculated reactivity from the chemically identified emissions. Total OH reactivity was measured using the Comparative Reactivity Method (CRM), and the chemical composition of the emissions was quantified with two gas chromatographs coupled to mass spectrometers (GC-MSs). Dynamic branch enclosures were used and emissions from one branch of a tree at the time were measured by rotating between them periodically.

Results show that birch had the highest values of total OH reactivity of the emissions (TOHRE), while pine had the lowest.

The main drivers for the known reactivity of pine and spruce were monoterpenes and sesquiterpenes. For birch, emissions were dominated by sesquiterpenes, even though monoterpenes and GLVs could be found too. However, calculated reactivity values remained low leading to the highest missing fraction of reactivity (>96 %), while pine and spruce had similar missing reactivity fractions between 56 % and 82 % (higher in the spring and decreasing as the summer proceeded). The high average values were driven by low reactivity periods and the fraction of missing reactivity got smaller for pine and spruce when the TOHRE values

increased. Important exceptions were identified for periods when the emission profiles changed from terpenes to Green Leaf Volatiles (GLVs), a family of compounds containing a 6 carbon atoms backbone with various functionalities (e.g. alcohols, aldehydes, esters) that indicate that the plant is suffering from stress. Then, very high TOHRE values were measured and the missing fraction remained high.

This study found a different trend in the missing OHRE fraction of Norway spruce from spring to autumn compared to one

previous study (Nölscher et al., 2013), which indicates that additional studies are required to fully understand the complexity of biogenic reactive emissions. Future studies of boreal trees in situ should be conducted to confirm the findings presented.

## 1 Introduction

The boreal forest is the largest continuous terrestrial biome and represents a third of the forested areas (Keenan et al., 2015). It is a large source of volatile organic compounds (VOCs), such as isoprene ($C_5H_8$), monoterpenes ($C_{10}H_{16}$), and sesquiterpenes





($C_{15}H_{24}$), as well as some light oxidized compounds such as methanol, acetaldehyde, and acetone (e.g. Lindfors and Laurila, 2000; Rinne et al., 2009). These compounds are emitted by vegetation and are therefore referred to as *biogenic* VOCs (BVOCs). Once in the atmosphere, these emissions undergo oxidation reactions by hydroxyl radical (OH), ozone ($O_3$), and nitrate radical ($NO_3$), and therefore influence the lifetime and concentrations of these oxidants.

OH is very reactive and therefore difficult to measure as well as to model (e.g. Heard and Pilling, 2003; Lelieveld et al.,
2016). Its concentrations show very high temporal and spatial variability. When observed OH concentrations are lower than predicted by global models, it is an indication of missing OH sinks in the models. As these sinks are most likely chemical sinks, Kovacs and Brune (2001) started to use total OH reactivity measurements as a tool to assess the exhaustiveness of chemical composition measurements of the atmosphere. This kind of measurements have been since performed in various environments (see the review by Yang et al., 2016) and Williams and Brune (2015) advocate for its widespread use at monitoring stations.
Based on these studies, Ferracci et al. (2018) modelled global OH reactivity to investigate the missing OH sinks.

By comparing the total OH reactivity with the reactivity derived from the known chemical composition of a sample, the gap in chemical composition knowledge can be identified. Particularly in forest environments, where these measurements have been made, this gap was found to be large. Di Carlo et al. (2004) observed first this missing reactivity at the Harvard forest station and this was later seen in other forests as well. Measurements of the total OH reactivity in a boreal forest, for instance,
using the comparative reactivity method (CRM, Sinha et al., 2008), have shown that less than half of the OH reactivity can be explained by the measured VOCs (Sinha et al., 2010; Nölscher et al., 2012).

The missing fraction (up to 89 % in Nölscher et al., 2012) for periods during which the forest experienced stressed conditions) is suspected to be the result of the incapacity to measure reactive compounds due to instrumental limitations. These compounds can be either VOCs directly emitted from the ecosystem (vegetation or soil) or oxidation compounds that are formed in the
45 atmosphere through oxidation reactions of these emitted compounds. However, Praplan et al. (2019) recently demonstrated that including modelled oxidation products of VOCs that are not measured is not sufficient to explain the missing reactivity at the site.

Therefore it becomes important to consider that the chemical composition of biogenic emissions have not been fully characterized. Applying total OH reactivity measurements to emissions allows to estimate in a similar fashion its unknown fraction (in
terms of reactivity). Previous measurements of the Total OH Reactivity of the Emissions (TOHRE) were contradictory. For instance, Kim et al. (2011) found for four tree species that their TOHRE matched the Calculated OH Reactivity of the Emissions (COHRE, calculated from individually quantified compounds in the emissions). However, these measurements were performed on very short time periods (< 24 h for each species). In contrast, Nölscher et al. (2013) found that while the TOHRE from Scots pine could be almost fully explained in the spring (15 % missing reactivity), TOHRE values were much higher than COHRE
in the summer (84 % missing reactivity) and in the autumn (70 % missing reactivity).

To further investigate the exhaustiveness of our knowledge on biogenic emissions and their specific influence on the observed missing OH reactivity, comprehensive, simultaneous VOC and OH reactivity measurements of emissions from three important boreal tree species were conducted at a boreal forest station, the second Station for Measuring Ecosystem-Atmosphere



Relations (SMEAR II) in Hyytiälä, Finland. The measurements alternated between seedlings of Scots pine (*Pinus sylvestris*),
Norway spruce (*Picea abies*), and downy birch (*Betula pubescens*) trees and lasted from May to October 2017.

## 2   Methods

### 2.1   Measurement site

Measurements were conducted at SMEAR II in Hyytiälä, Finland, (61°51' N, 24°17' E, 181 m above sea level; see Hari and
Kulmala, 2005), about 60 km North-East from the city of Tampere. The station is located in a ca. 60-year old managed mixed
conifer forest dominated by Scots pine (*Pinus sylvestris*) homogeneously for about 200 m in all directions from its mast, which
carries instrumentation for various observations. This data as well as additional data acquired at the site are available via the
Smart-SMEAR portal (https://avaa.tdata.fi/web/smart/smear/search; Junninen et al., 2009).

For this study, the measurements were done at a container located next to an opening about 115 m south from the mast. The
instrumentation to measure VOC emissions (section 2.4) and TOHRE (section 2.5.1) was located inside the container. The
seedlings used in this study (section 2.2) were located just outside of the container and received direct sunlight for most of the
day. Branch enclosures (section 2.3) were used to investigate their emissions.

### 2.2   Seedlings

Seedlings for each of the studied tree species - Scots pine (*Pinus sylvestris*), Norway spruce (*Picea abies*), and downy birch
(*Betula pubescens*) - were brought from a commercial nursery (Harviala Oy, Harviala, Finland) to the site. The seedlings were
100–150 cm tall and were planted in 10 L plastic pots in a mixture of sand and peat. They were watered regularly.

For each tree, the enclosure was moved to a different branch twice during the campaign. Each time this occurred and at the
end of the last measurement period, the branch from which emissions were measured last was cut in order to determine the dry
weight of the needles' or leaves' biomass for three periods for each tree. To do so, the needles or leaves from the cut branches
were dried at 80°C overnight and subsequently weighed. Dry weights of the needles or leaves of the different branches can be
found in Table B1 of the Appendix.

No correction for the growth of the biomass was applied during the growth period (May-June) as the cutting of the branches
happened in general right after the measurement period so that it can be assumed that the changes in biomass remain small in
comparison to other uncertainties of total OH reactivity measurements.

### 2.3   Dynamic branch enclosures

Hakola et al. (2006) describe in detail the method used. Briefly, the enclosure consists of a ca. 6 L-cylinder made of transparent
Teflon, which is attached to the branch on one side and to a Teflon frame equipped with inlet and outlet ports on the other
side. VOC-free air provided by a generator (HPZA-7000, Parker Balston, Lancaster, NY, U.S.A.) flow through the enclosure
at about $4\,\mathrm{L\,min^{-1}}$ (flow $f$). The relative humidity (RH) and the temperature in the enclosure were recorded with a thermistor





(Philips KTY 80/110, Royal Philips Electronics, Amsterdam, Netherlands) and the Photosynthetically Active Radiation (PAR)
was measured with a quantum sensor (LI-190SZ, LI-COR, Biosciences, Lincoln, USA) placed on top of the enclosure frame.

In this study, three branch enclosures were used so that they could be set up one or two weeks before the measurements of
the emissions in order to reduce the stress caused by handling the branches to a minimum. During that time, the enclosure was
left open and only when the measurement started, the enclosure was carefully closed, with transparent Teflon film, which could
nevertheless result in some level of stress.

The temperature difference between ambient conditions and inside the enclosure are presented in the Appendix (Figure C1).
For a large majority of the data (74 %), the difference lies within 3°C. For another 22 % of the data the difference is comprised
between 3 and 10°C. The maximum temperature difference is 27.5°C. Large temperature differences happened when prolonged
direct sunlight heated up the enclosure.

## 2.4  In-situ measurements of Volatile Organic Compounds

Volatile Organic Compounds (VOCs) were measured with two in situ GC-MSs, which have been previously described in
more detail by Hellén et al. (2017, 2018). One GC-MS measured the concentrations of mono- and sesquiterpenes, isoprene,
2-methyl-3-butenol (MBO) and $C_{5-10}$ aldehydes in the emissions. These compounds were collected for 30 minutes from a
$40\,\mathrm{ml\,min^{-1}}$ subsample flow of the CRM instrument sampling flow in the cold trap (Carbopack B/Tenax TA) of the thermal
desorption unit (TurboMatrix, 650, Perkin-Elmer) connected to the GC (Clarus 680, Perkin-Elmer) coupled to the MS (Clarus
SQ 8 T, Perkin-Elmer). A HP-5 column (60m, i.d. 0.25 mm, film thickness 1 μm) was used for separation.

The other GC-MS measured the concentrations of alcohols and volatile organic acids (VOAs). Every other hour a sample
was taken for 60 minutes and analysed with a thermal desorption unit (Unity 2 + Air Server 2, Markes International LTD,
Llantrisant, UK) connected to the GC (Agilent 7890A, Agilent Technologies, Santa Clara, CA, USA) and the MS (Agilent
5975C, Agilent Technologies, Santa Clara, CA, USA). A polyethylene glycol column DB-WAXetr (30-m, i.d. 0.25 mm, a film
thickness 0.25 μm) was used for the separation.

## 2.5  OH reactivity

The OH reactivity is the inverse of the OH lifetime. OH reactivity, $R_{\mathrm{OH}}$ can be calculated from the sum of the concentration
of individually emitted compounds $X_i$, $[X_i]$, multiplied by their respective reaction rate coefficient with OH ($k_{\mathrm{OH}+X_i}$):

$$R_{\mathrm{OH}} = \sum_i [X_i] k_{OH+X_i} \tag{1}$$

The experimental total OH reactivity, $R_{\mathrm{exp}}$, can be measured with the Comparative Reactivity Method (CRM, Sinha et al.,
2008; Michoud et al., 2015). The specific instrument used for this study is described in Praplan et al. (2017, 2019) and the
measurement principle briefly explained in the following section together with the application of the method to measure the
OH Reactivity of Emissions (OHRE).





### 2.5.1 Total OH reactivity measurements: the Comparative Reactivity Method

The CRM is based on monitoring the signal change of pyrrole ($C_4H_5N$) exposed to OH in a reactor together with either clean (*zero*) air or air sampled from the branch enclosure. OH is produced by the photolysis of water ($H_2O$) in a nitrogen flow (99.9999% $N_2$) using ultraviolet (UV) radiation and a gas chromatograph (GC, SYNTECH SPECTRAS Analyser GC955, Synspec BV, Groningen, The Netherlands) equipped with a photon ionization detector (PID) measures the pyrrole concentration in the CRM instrument reactor every two minutes. Based on pyrrole calibrations for the GC-PID detector, a sensitivity of

1678 $\text{ppb}_v^{-1}$ measured on 11 May was used for data until 14 June, then a sensitivity of 1833 $_v^{-1}$ measured on 15 June was used for data until 28 June. On 28 June, a lower sensitivity of 1193 $_v^{-1}$ was measured and used for rest of the measurement periods.

During zero air measurements all OH is consumed by pyrrole (labelled $C_2$ level). This zero air is produced by passing the sampled air through a platinum catalyst heated at ca. 450 °C to remove reactive species. When zero air is replaced with the sampled air other reactive compounds compete for OH, leading to an increased pyrrole concentration ($C_3$ level). The instrument

alternates measurements of zero air and sampled air every 8 minutes. The conditions in the reactor after switching stabilize within one minute and therefore the first pyrrole measurement after each switch is discarded. The amount of pyrrole in the reactor in the absence of OH with the UV light on ($C_1$ level) is slightly lower than the amount of pyrrole introduced into the reactor in the dark ($C_0$ level), due to photolysis of pyrrole (5.6–9.3 %). $C_1$ is measured by introducing a large concentration of a 0.6 % propane ($C_3H_8$) in nitrogen ($N_2$) gas mixture to act as an OH scavenger (Zannoni et al., 2015). From the difference

between $C_2$ and $C_3$ pyrrole levels and taking into account the amount of available pyrrole ($C_1$), the total OH reactivity in the reactor $R_{\text{eqn}}$ can be derived from the following equation:

$$R_{\text{eqn}} = \frac{C_3 - C_2}{C_1 - C_3} \cdot k_p \cdot C_1 \tag{2}$$

with $k_p$ the reaction rate of pyrrole with OH ($1.2 \cdot 10^{-10}$ cm$^3$ s$^{-1}$, Atkinson et al., 1985). However, this equation has been derived under a pseudo first-order kinetics assumption (i.e. [$C_4H_5N$]$>>$[OH]), but the pyrrole-to-OH ratio (pyr:OH) varies

between 1.0 and 3.5 in the present study.

Therefore we apply a correction described in detail in (Praplan et al., 2019) for this deviation from pseudo first-order kinetics, based on experimental reactivity calibrations with $\alpha$-pinene (see section 2.5.3). The only difference here compared to Praplan et al. (2019) who applied the correction factor to ambient measurements is that the background reactivity of the empty enclosure ($R_{\text{eqn,blank}}$) is also taken into account. $R_{\text{eqn,blank}}$ was determined between 28 September and 4 October and is $2.6 \pm 3.0$ s$^{-1}$

($1\sigma$, see Fig. C2 in the Appendix). Based on this, the reactivity in the reactor ($R_{\text{CRM}}$) is derived according to the following equation:

$$R_{\text{CRM}} = ((R_{\text{eqn}} - R_{\text{eqn,blank}}) + 0.449)/0.497 \tag{3}$$





In addition, because of the dilution of the sampled air with humid nitrogen, the calculation of the total OH reactivity of the sampled air $R_{\mathrm{exp}}$ requires the use of the dilution factor $D$ (ratio of sampling flow over total flow through the reactor, comprised between 0.63 and 0.69):

$$R_{\mathrm{exp}} = R_{\mathrm{CRM}}/D \qquad (4)$$

Other correction factors need to be applied during CRM data analysis. However, corrections due to the presence of ozone ($O_3$) and nitrogen oxides ($NO_x$) described elsewhere (e.g. Michoud et al., 2015; Fuchs et al., 2017; Praplan et al., 2017, 2019) are not required in the present study due to the use of zero air through the dynamic branch enclosure. Only the correction due to the difference in relative humidity (RH) in the reactor between $C_2$ and $C_3$ levels and the correction due to the deviation from pseudo-first-order kinetics need to be taken into account. A detailed description of these corrections can be found in the following subsections.

Finally, the Total OH Reactivity of Emissions (TOHRE) measured using a dynamic branch enclosure can be derived from

$$\mathrm{TOHRE} = R_{\mathrm{exp}} \cdot f/m_{\mathrm{dw}} \qquad (5)$$

where $f$ is the total flow through the enclosure and $m_{\mathrm{dw}}$ is the dry weight of the leaves or needles in the enclosure. In a similar way, the Calculated OH Reactivity of Emissions (COHRE), based on the known air composition, can be calculated.

$$\mathrm{COHRE} = R_{\mathrm{OH}} \cdot f/m_{\mathrm{dw}} \qquad (6)$$

### 2.5.2 Correction due to the difference in RH

Equation (2) assumes that RH (i.e. OH levels) are identical during $C_2$ and $C_3$ measurements. In order to minimize the difference of RH between $C_2$ and $C_3$, zero air is produced in the CRM instrument by passing sampled air from the dynamic branch enclosure through a catalytic converter (1 % wt. platinum on aluminium oxide pellets, Sigma-Aldrich, Co., St. Louis, MO, USA), which remove VOCs, but does not affect RH levels much. However, the small decrease in RH after the catalytic converted for $C_2$ measurements needs t o be taken into account. Figure 1 show the pyrrole signal as a function of RH while measuring zero air. The applied correction is then:

$$C_2 = C_{2,\mathrm{uncorrected}} - 0.088 \cdot (\mathrm{RH}_{C_3} - \mathrm{RH}_{C_2}) \qquad (7)$$

### 2.5.3 Correction due to deviation from pseudo first-order kinetics

As mentioned previously, this correction is necessary as Eq. (2) is derived under the assumption of a pseudo-first-order kinetics ($[C_4H_5N] \gg [OH]$), while the experimental pyrrole-to-OH ratio (pyr:OH) is comprised between 1.0 and 3.5. Originally, Sinha




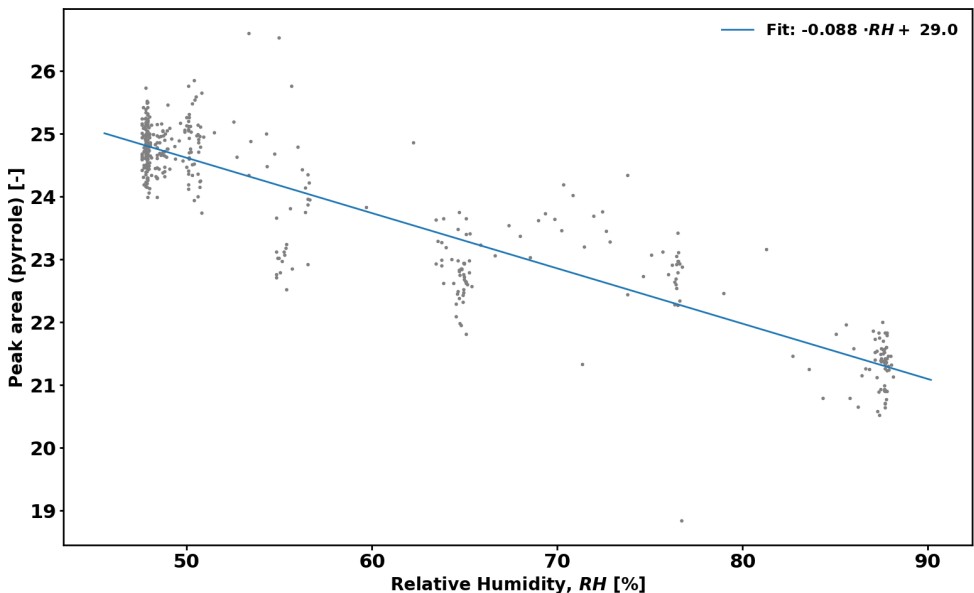

**Figure 1.** Change in pyrrole level ($C_2$) according to relative humidity (RH) in the CRM reactor.

et al. (2008) used a very simple two-equation model for this correction. Michoud et al. (2015) opted for an empirical approach
based on experimental calibration using gas standards, as they demonstrated that the model was not reproducing accurately
the observed response of pyrrole in the reactor, despite alterations to take into account secondary OH chemistry. In the present
study we use the experimental results derived in Praplan et al. (2019) based on $\alpha$-pinene calibrations, which show that the
measured OH reactivity ($R_{\mathrm{eqn}}$) is roughly half the expected reactivity, so that the exact relationship between the reactivity in
the reactor ($R_{\mathrm{CRM}}$) and $R_{\mathrm{eqn}}$ is the following:

$$R_{\mathrm{CRM}} = ((R_{\mathrm{eqn}} - R_{\mathrm{eqn,blank}})) + 0.449) \cdot 0.497 \tag{8}$$

### 2.6 Emission models

We used a typical model for VOC emissions (Guenther et al., 1993, 1995) to test the light and temperature dependence of
TOHRE. The temperature-only dependence is the same dependence as the one for monoterpene emissions and is expressed
with the following equation:

$$\mathrm{TOHRE} = \mathrm{TOHRE}_S \cdot \exp[\beta(T - T_S)] \tag{9}$$





TOHRE$_S$ is the TOHRE at standard temperature $T_S$ (303 K) and $T$ the temperature in the enclosure. In the present study, we assume that the leaf surface temperature is the same as the temperature inside the enclosure. $\beta$ describes the temperature dependence (so-called $\beta$-factor) and is estimated to be $0.09\,\mathrm{K}^{-1}$ for monoterpenes.

A hybrid algorithm based on both temperature and light can be used to model emissions which follow also changes in illumination (Guenther, 1997; Ghirardo et al., 2010). The dependence on light and temperature for TOHRE is then formulated as follow:

$$\mathrm{TOHRE} = \mathrm{TOHRE}_{0,\mathrm{pool}} \cdot \exp[\beta(T - T_S)] + \mathrm{TOHRE}_{0,\mathrm{synth}} \cdot c_L \cdot c_T \tag{10}$$

with $\mathrm{TOHRE}_{0,\mathrm{pool}}$ and $\mathrm{TOHRE}_{0,\mathrm{synth}}$ the standard TOHRE potentials pool emissions (stored compounds, temperature dependent) and synthesis emissions (newly synthesised compounds, light and temperature dependent), respectively. $c_L$ and $c_T$ are light and temperature activity coefficients, respectively, defined such as:

$$c_L = \frac{\alpha c_{L1} Q}{\sqrt{1 + \alpha^2 Q^2}} \tag{11}$$

$$c_T = \frac{\exp\left(\frac{c_{T1}(T - T_S)}{R T_S T}\right)}{1 + \exp\left(\frac{c_{T2}(T - T_M)}{R T_S T}\right)} \tag{12}$$

$T$ and $T_S$ are the same as above and $Q$ is the PAR measured just above the enclosure. $\alpha$ (0.0027), $c_{L1}$ (1.066), $c_{T1}$ (95000 mol $J^{-1}$), $c_{T2}$ (230000 mol $J^{-1}$), and $T_M$ (314 K) are empirical coefficients. Finally, $R$ is the gas constant (8.314 J $K^{-1}$ mol$^{-1}$).

## 3 Results and discussion

### 3.1 Overview

An overview of monthly averages for TOHRE and missing TOHRE (absolute and fraction) can be found in Table 1. The highest TOHRE monthly averages were found for birch in May and June ($1.6$–$2.6\,10^{-3}\,\mathrm{m}^3\,\mathrm{s}^{-2}\,\mathrm{g}_{\mathrm{dw}}^{-1}$), which is mostly unaccounted for (missing OHRE fraction 96–99 %). The monthly TOHRE averages from spruce were high in July and August ($1.1$–$1.5\,10^{-3}\,\mathrm{m}^3$ $\mathrm{s}^{-2}\,\mathrm{g}_{\mathrm{dw}}^{-1}$), while the highest monthly average for TOHRE from pine was in July ($6.1\,10^{-4}\,\mathrm{m}^3\,\mathrm{s}^{-2}\,\mathrm{g}_{\mathrm{dw}}^{-1}$). A few compounds per class of biogenic VOCs were identified as the main drivers of the reactivity and this will be discussed in the following subsections for each tree individually.

The results illustrate as well how biogenic reactivity is influenced by the time of the year and the tree species found in the forested areas. In addition, high measured TOHRE is related to a change in the emission profiles with a larger fraction of Green Leaf Volatiles (GLVs). GLVs form a family of $C_6$ compounds, including aldehydes, alcohols and esters, which are emitted rapidly and in large amount during stress periods (e. g. Scala et al., 2013). Stress can have various abiotic and biotic causes (e. g. drought, attack by pathogens or herbivores).





**Table 1.** Monthly averages of temperature in the enclosure ($T_e$), relative humidity (RH), Photosynthetically Active Radiation (PAR) and Total OH Reactivity of the Emissions (TOHRE), as well as missing OHRE (absolute and relative). The number of observations, $n$, for missing OHRE is lower than for other parameters due to incomplete overlap between calculated OHRE (VOC data) and TOHRE.

| | $n_{days}$ | $T_e$ | RH | PAR | TOHRE | Missing OHRE | Missing OHRE |
|---|---|---|---|---|---|---|---|
| | | [°C] | [%] | [µmol m$^{-2}$ s$^{-1}$] | [m$^3$ s$^{-2}$ g$_{dw}^{-1}$] | [m$^3$ s$^{-2}$ g$_{dw}^{-1}$] | (fraction) |
| **Pine** | | | | | | | |
| June | 10 ($n$=753) | 15.6 ± 6.0 | 20.6 ± 4.8 | 90 ± 175 | 9.6 ± 11.2 · 10$^{-5}$ | 7.6 ± 8.0 · 10$^{-5}$ ($n$=727) | 0.77 ± 0.26 |
| July | 8 ($n$=542) | 15.5 ± 5.2 | 22.8 ± 7.0 | 71 ± 138 | 6.1 ± 6.2 · 10$^{-4}$ | 5.3 ± 5.4 · 10$^{-4}$ ($n$=506) | 0.78 ± 0.17 |
| August | 7 ($n$=535) | 15.9 ± 4.8 | 19.5 ± 3.0 | 46 ± 84 | 1.8 ± 1.8 · 10$^{-4}$ | 1.4 ± 1.3 · 10$^{-5}$ ($n$=364) | 0.59 ± 0.31 |
| September | 8 ($n$=621) | 8.8 ± 2.2 | 39.6 ± 8.4 | 30 ± 42 | < l.o.d. | - | - |
| **Spruce** | | | | | | | |
| May | 10 ($n$=664) | 13.2 ± 10.3 | 25.8 ± 7.6 | 24 ± 41 | 2.5 ± 1.5 · 10$^{-4}$ | 2.0 ± 1.3 · 10$^{-4}$ ($n$=458) | 0.82 ± 0.22 |
| June | 0 ($n$=0) | - | - | - | - | - | - |
| July | 9 ($n$=708) | 16.0 ± 6.5 | 16.2 ± 3.5 | 13 ± 28 | 1.5 ± 4.1 · 10$^{-3}$ | 7.9 ± 29.5 · 10$^{-4}$ ($n$=658) | 0.56 ± 0.25 |
| August | 8 ($n$=625) | 16.3 ± 3.4 | 17.2 ± 4.7 | 54 ± 68 | 1.1 ± 1.7 · 10$^{-3}$ | 9.8 ± 15.6 · 10$^{-4}$ ($n$=604) | 0.58 ± 0.33 |
| **Birch** | | | | | | | |
| May | 8 ($n$=671) | 13.4 ± 5.8 | 22.2 ± 4.5 | 30 ± 30 | 2.6 ± 1.4 · 10$^{-3}$ | 2.5 ± 0.6 · 10$^{-3}$ ($n$=582) | 0.99 ± 0.02 |
| June | 15 ($n$=1133) | 11.9 ± 6.9 | 29.1 ± 4.5 | 17 ± 34 | 1.6 ± 0.9 · 10$^{-3}$ | 1.5 ± 0.9 · 10$^{-3}$ ($n$=980) | 0.96 ± 0.15 |
| July | 7 ($n$= 533) | 15.9 ± 8.3 | 25.7 ± 5.5 | 14 ± 31 | 6.8 ± 6.3 · 10$^{-4}$ | 6.4 ± 5.4 · 10$^{-4}$ ($n$=506) | 0.84 ± 0.29 |

In general the missing OHRE fraction was higher in spring and decreased as the seasons proceeded. Missing OHRE fraction
from birch remained high from May to July (99–84 %), making it the least understood reactivity. Pine and spruce had similar
fractions of missing OHRE (59–78 % and 56–82 %, respectively). These fractions are partly due to uncertainties both on the
measured TOHRE and COHRE. For TOHRE, the correction for deviation from pseudo first-order kinetics applied to CRM
data is based on calibration with $\alpha$-pinene as a surrogate for biogenic emissions, but monoterpenes do not always represent the
largest fraction of the emissions, which result in some uncertainty in TOHRE. On the other hand, unidentified sesquiterpenes
have been found in emissions from all three tree species (see Appendix D) and their quantification was performed using
surrogates and their reaction rates was assumed to be an average one, based on the reaction rates for other sesquiterpenes
($10^{-10}$ cm$^3$s$^{-1}$). This also introduce some uncertainty. Notwithstanding these uncertainties it will appear in the following
discussion that high missing OHRE values averages are driven by low reactivity values and measurement scatter. When TOHRE
is clearly above the background values, the missing fraction is reduced, which indicates a generally good understanding of
emissions with the exception of periods dominated by GLVs. There TOHRE values are high but the missing fraction also
remains high and this cannot be explain only by measurement and calculation uncertainties.

## 3.2 Pine

The known OH reactivity for pine emissions is dominated by monoterpenes with a small fraction of sesquiterpenes as expected
from earlier studies (Tarvainen et al., 2005; Hakola et al., 2006; Yassaa et al., 2012; Bäck et al., 2012; Faiola et al., 2018).





Branches were cut on 15 June, 16 Augsut, and 11 October, but emission profiles from all three branches from this same seedling are similar. Here COHRE is mostly driven by $\alpha$-pinene, limonene, and $\Delta^3$-carene. Sesquiterpenes (mostly $\alpha$- and $\beta$-farnesene) contribute up to 15 % to the known OH reactivity and MBO represent an important fraction, especially in June and July. TOHRE qualitatively follows COHRE.

   The highest TOHRE values from pine were measured in early July and early October. These two periods (3–5 July and 4–11

October) are marked with a fraction of Green Leaf Volatiles (GLVs) up to roughly 35 % (mostly due to *cis*-3-hexenol). At the same time, emissions from monoterpenes previously mentioned as well as terpinolene increase as well. Between 3 and 5 July TOHRE increased and was high even at night, when it is otherwise close to zero at night. Interestingly 3 July marks the end of a warm and sunny period with the maximum temperature in the branch enclosure between 30 and 40°C for 5 days in a row and the beginning of a cooler and cloudier period with some precipitations. It is not clear though if stress emissions are related

to the change in environmental conditions or are a result of stress experienced during the previous days.

   For the other periods, TOHRE follows COHRE quantitatively but is usually higher than it. Only in September the missing fraction is the lowest, due to the low TOHRE values measured, which are in the same range as the COHRE values (only with a much larger scatter).

   Nölscher et al. (2012) found higher missing reactivity for ambient measurement at SMEAR II, a boreal site dominated by

245 Scots Pine, for stress periods (elevated temperature). While here the stress emissions were not due to elevated temperature (see section3.5), the missing OHRE was generally higher during these periods, indicating that some of these stress-related emissions are not terpenoids or oxidized volatile organic compounds.





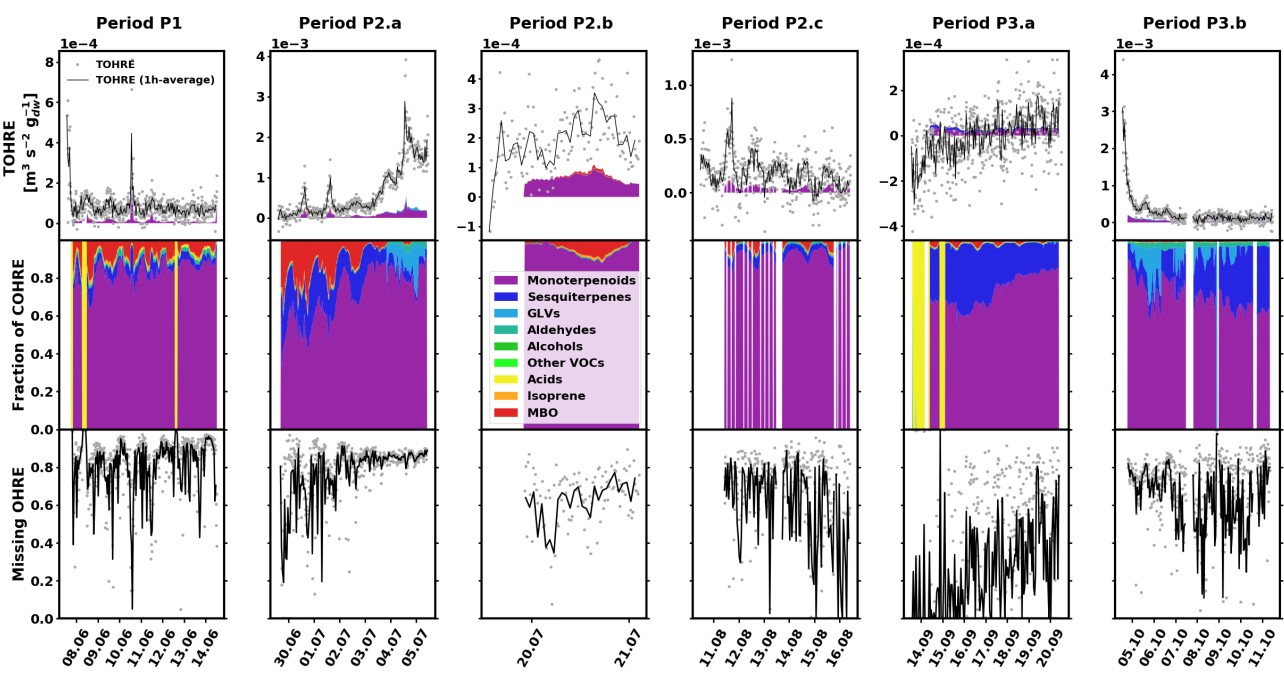

**Figure 2.** Measured Total OH Reactivity of the Emissions (TOHRE),its 1h-average values, and COHRE (coloured surface) normalized to dry weight (top row) for pine. Fraction of the various contributions of chemical species to COHRE (middle row). Note that the same colour code is used for COHRE in the top row. Missing fraction of OHRE (bottom row). The periods are denoted "P" for "pine", the number indicates which branch is being measured and the letter indicate which period it is for the same branch.





### 3.3 Spruce

For spruce TOHRE follows qualitatively COHRE as well. Branches were cut on 21 June, 9 August, and 5 November. The
250 known reactivity of the emissions in May (first branch) is dominated by monoterpenes, which is expected from earlier studies
(Yassaa et al., 2012; Hakola et al., 2017; Wang et al., 2017). The main drivers are limonene, $\beta$-pinene, and $\beta$-phellandrene.

The highest TOHRE values is observed at the beginning of July (for the second branch) with one extremely high peak
over $0.06\,\mathrm{m}^3\,\mathrm{s}^{-2}\,\mathrm{g}_{dw}^{-1}$ on 9 July and another TOHRE peak on the next day. However, almost all reactivity can be explained by
monoterpenes and GLVs during that period (mostly *cis*-3-hexen-1-ol and *cis*-3-hexenylacetate, as well as limonene). 9 and 10
July were dry and sunny days, with mximum temperatures in the branch enclosure close to $40\,^{\circ}\mathrm{C}$. After that, when the weather
gets cooler and cloudier with some precipitations between 11 and 14 July, the GLV fraction decreases and monoterpenes and
sesquiterpenes are accounting for most of the known reactivity. This is in stark contrast with the observed stress emissions
from pine in this study, which increased during the colder period, after a warm spell.

However, between 19 and 23 August (third branch) high TOHRE values (up to $0.01\,\mathrm{m}^3\,\mathrm{s}^{-2}\,\mathrm{g}_{dw}^{-1}$) were measured (including
260 at night), similarly to the stress period observed for pine. It can be seen that during these periods with larger fraction of GLVs
some needles were drying and falling (Appendix A), which confirms that the tree suffered stress (most probably drought).
Other environmental conditions did not change much during that period, which was relatively cool and cloudy.

In contrast to stress periods in pine, monoterpene emissions from spruce were low when the GLV fraction increased. Dur-
ing this period, *cis*-3-hexen-1-ol, *cis*-3-hexenylacetate, and *trans*-2-hexenal mostly contribute to COHRE. In September, this
branch had low TOHRE and the known reactivity of the emissions was caused by monoterpenes and sesquiterpenes, similarly
to the period between 16 and 19 August, before the large stress episode. $\alpha$-Farnesene contributes most to the sesquiterpene re-
activity fraction (here and for other periods as well). The increase of the sesquiterpene fraction in the emissions is in agreement
with observations from Hakola et al. (2017) (up to $75\,\%$ of the emissions in late summer, mostly $\beta$-farnesene).

A direct comparison with the results for TOHRE and missing OHRE of spruce from Nölscher et al. (2013) is difficult due
to the many factors affecting the emissions. While they found that the missing OHRE was lower in the spring and increased in
the late summer and autumn to $70$–$84\,\%$, the present study suggest that the missing OHRE fraction is decreasing from May to
August. As discussed earlier, lots of high missing OHRE in the present study stem from low reactivity periods with high scatter
for TOHRE and values close to zero for COHRE. However, because Nölscher et al. (2013) assume a constant emission profile
(measured in spring) throughout the year and otherwise rely on PTR-MS data (without speciation), it is imaginable that the
275 chemical compositions of the emissions changed with the season to more reactive monoterpenoids or sesquiterpenes, leading
to an underestimation of the calculated OH reactivity.



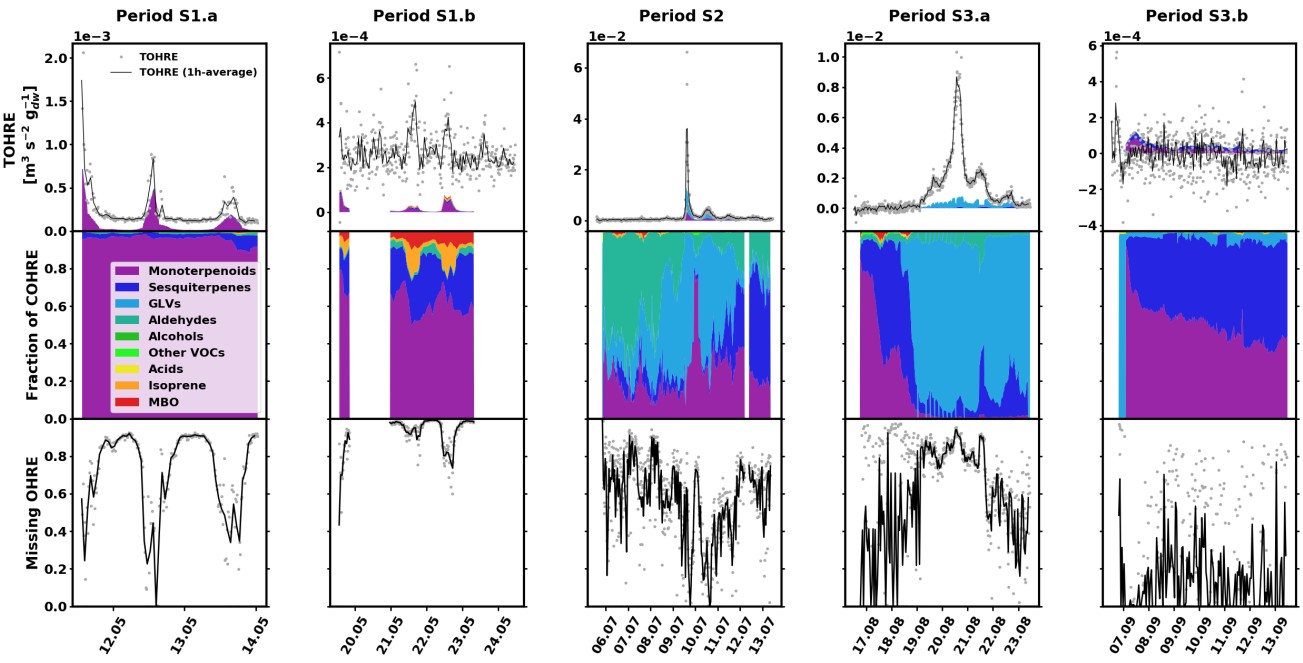

**Figure 3.** Measured Total OH Reactivity of the Emissions (TOHRE),its 1h-average values, and COHRE (coloured surface) normalized to dry weight (top row) for spruce. Fraction of the various contributions of chemical species to COHRE (middle row). Note that the same colour code is used for COHRE in the top row. Missing fraction of OHRE (bottom row). The periods are denoted "S" for "spruce", the number indicates which branch is being measured and the letter indicate which period it is for the same branch.





### 3.4 Birch

Birch branches were cut on 21 June, 9 August and 6 September. The observed TOHRE shows relatively high values (due to the low dry weight mass) with almost no diurnal pattern. In late June a weak pattern can be observed and in mid-July a few

reactivity peaks can be observed (second branch). It is possible that the constant blank value subtracted from the measurements underestimates at time the actual background of the measurements.

Here TOHRE follows COHRE quantitatively once more, but the missing fraction of OHRE is consistently high. This is partly due to the generally low values of COHRE, which is dominated by sesquiterpenes for the first two branches (until 9 August), with a significant amount of monoterpenes (up to 40 %). Periods when the known reactivity is dominated by organic

acids are missing terpene measurements. In May for the first branch $\beta$-caryophyllene, $\alpha$-humulene, and another unidentified sesquiterpene, as well as sometimes cis-3-hexenylacetate contribute most to the reactivity of the emissions. For the second branch in June and July, the emission profile is slightly different with $\beta$-caryophyllene, $\alpha$-farnesene, and linalool as well as sometimes *cis*-3-hexenylacetate and *cis*-3-hexen-1-ol (co-emitted) contributing most.

For the last branch measured in August, a significant fraction (up to 50 %) of the known reactivity comes from GLVs

(again *cis*-3-hexenylacetate and *cis*-3-hexen-1-ol), but the fraction of sesquiterpenes (mostly $\alpha$-farnesene) is smaller while monoterpenes (carene, $\alpha$-pinene and $\alpha$-terpineol) contribute more. Pictures in Appendix A show how some leaves turned brown, possibly indicating the end of the growing season and the senescence of the leaves.

Haapanala et al. (2009) found a large fraction of $\alpha$-farnesene in mountain birch emissions on a given year, but stressed that there was an important inter-annual variation in the emission profile, with almost no $\alpha$-farnesene detected on the following

295 year for the same branch.



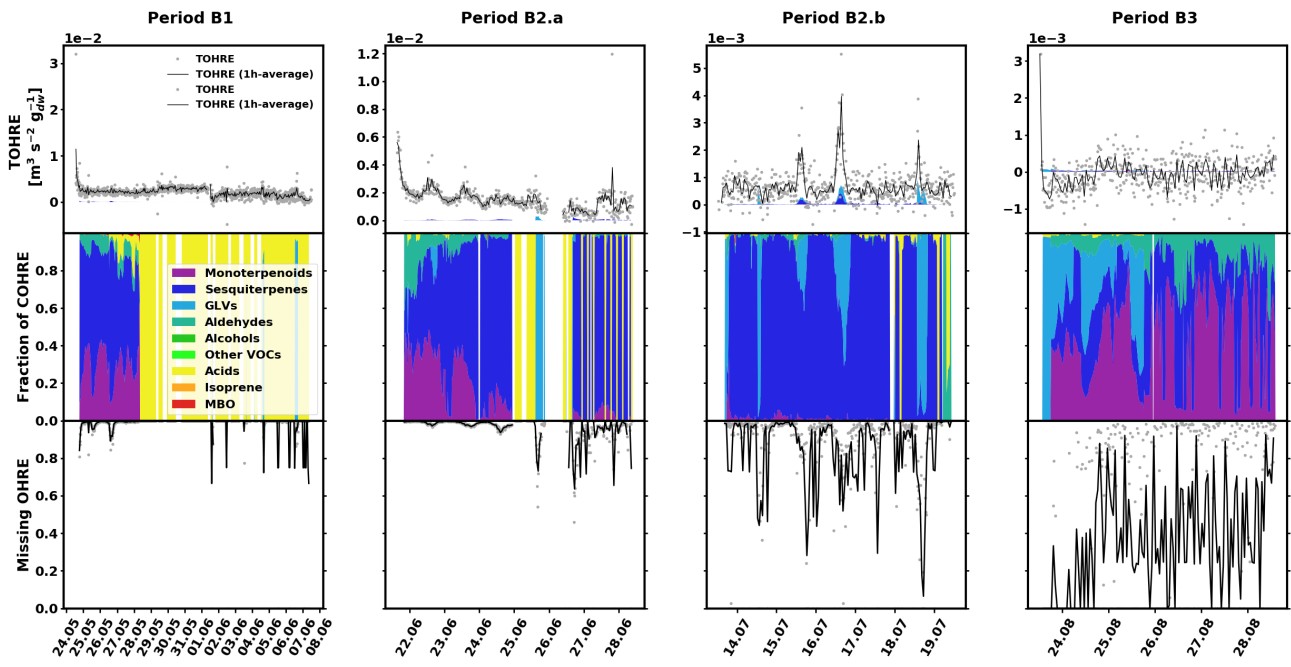

**Figure 4.** Measured Total OH Reactivity of the Emissions (TOHRE),its 1h-average values, and COHRE (coloured surface) normalized to dry weight (top row) for birch. Fraction of the various contributions of chemical species to COHRE (middle row). Note that the same colour code is used for COHRE in the top row. Missing fraction of OHRE (bottom row). The periods are denoted "B" for "birch", the number indicates which branch is being measured and the letter indicate which period it is for the same branch.





### 3.5 Temperature and light dependence of TOHRE

In order to also study the dependence of TOHRE on temperature, TOHRE has been plotted against the temperature in the enclosure and regressions derived from Eq. (9) have been performed (Fig. 5 and Table 2). Similar figures for COHRE, and missing OHRE can be found in Appendix E showing similar findings than for the TOHRE dependence on temperature.

Good correlations with temperature are found for the TOHRE of pine in June and August ($R = 0.70$ and $0.61$, respectively), in May and July for spruce ($R = 0.59$–$0.50$) and in July for birch ($R = 0.71$). Periods with no correlation were connected to either stress events (in particular July for pine) or low TOHRE values with no diurnal variations (as in September for pine and spruce and in May-June for birch). Because of this, averaging the whole dataset leads to low coefficients of correlation ($R = 0.23$–$0.37$).

Considering values of $\beta$-factors from monthly regressions with $R > 0.5$, they range from $0.0246$ to $0.1853\,°\mathrm{C}^{-1}$. Guenther et al. (2012) recommended a value of $0.10°\mathrm{C}^{-1}$ to model monoterpene emissions. For sesquiterpenes, average values ranging from $0.14$ to $0.22$ have been reported (e. g. Tarvainen et al., 2005; Hakola et al., 2006; Duhl et al., 2008), even though values as low as $0.025$, $0.05$ and $0.056$ were found as well (Tarvainen et al., 2005; Helmig et al., 2007; Ruuskanen et al., 2007, respectively). For pine, which is dominated by monoterpene emissions, $\beta$-factors are about $0.09$–$0.10°\mathrm{C}^{-1}$ except for stress
periods, when the $\beta$-factor is then smaller than $0.003$. For spruce, $\beta$-factors increase from $0.02$ to $0.19°\mathrm{C}^{-1}$ between May and July, demonstrating a clear regime change in the temperature dependence of the emissions, with an increasing contribution of less volatile compounds (sesquiterpenes and GLVs). For birch, the $\beta$-factor in July when a good correlation with temperature was found remains low, even though emissions are dominated by sesquiterpenes. This might be an indication of emissions of non-terpenoid volatile compounds.

Results of using Eq. (10) to include the effect of light on TOHRE (Hybrid algorithm, Table 2) show that in general only small improvements (increases of $R$) are achieved. In a few cases $R$ was even slightly reduced. One notable exception is a large improvement of the coefficient of correlation $R$ from $0.5$ to $0.9$ for spruce in July. The addition of a small $\mathrm{TOHRE}_{0,\mathrm{synth}}$ term seemed to be enough to capture the large peak that was reported as stress, indicating a radiation-induced stress in this case.

$\beta$-Factors are in general very similar to the results of the regression for the temperature-only dependence (when a good
correlation was found in the first place). Note that in September (and to some extend in August) the temperature range remain small (about $10\,\mathrm{K}$) and towards small temperature, so that nothing conclusive can be inferred from these results. In summary the effect of light on reactive remissions remains limited in the present study, but other factors can play a major role on the type and amount of reactive emissions.





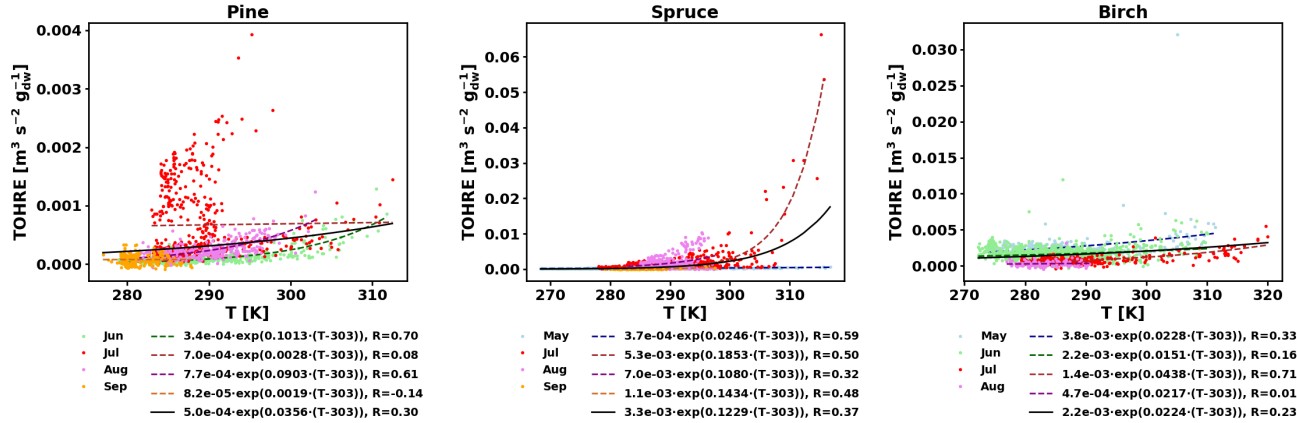

**Figure 5.** TOHRE as a function of temperature in the branch enclosure for pine (left), spruce (center), and birch (right). Coloured dots and dashed lines are data separated by month (data and exponential regression) and the solid black line is the exponential regression for all data.





**Table 2.** Regression coefficients and coefficients of correlation ($R$) for temperature dependence of TOHRE and for its dependence on both temperature and light using the hybrid algorithm.

| | Temperature dependence | | | Hybrid algorithm | | | |
|---|---|---|---|---|---|---|---|
| | $\text{TOHRE}_s$ | $\beta$ | $R$ | $\text{TOHRE}_{0,\text{pool}}$ | $\beta$ | $\text{TOHRE}_{0,\text{synth}}$ | $R$ |
| | [$m^3$ $s^{-2}$ $g_{dw}^{-1}$] | [$K^{-1}$] | | [$m^3$ $s^{-2}$ $g_{dw}^{-1}$] | [$K^{-1}$] | [$m^3$ $s^{-2}$ $g_{dw}^{-1}$] | |
| **Pine** | | | | | | | |
| June | 3.4e-04 | 0.1013 | 0.70 | 3.4e-04 | 0.1013 | 2.0e-07 | 0.79 |
| July | 7.0e-04 | 0.0028 | 0.08 | 7.0e-04 | 0.0028 | 4.0e-11 | 0.02 |
| August | 7.7e-04 | 0.0903 | 0.61 | 7.7e-04 | 0.0903 | 1.8e-10 | 0.67 |
| September | 8.2e-05 | 0.0019 | -0.14 | 7.9e-05 | 0.0000 | 4.3e-10 | -0.14 |
| All | 5.0e-04 | 0.0356 | 0.30 | 5.0e-04 | 0.0356 | 6.8e-19 | 0.17 |
| **Spruce** | | | | | | | |
| May | 3.7e-04 | 0.0246 | 0.59 | 3.4e-04 | 0.0207 | 1.3e-02 | 0.47 |
| July | 5.3e-03 | 0.1853 | 0.50 | 5.3e-03 | 0.1853 | 2.1e-20 | 0.90 |
| August | 7.0e-03 | 0.1080 | 0.32 | 7.0e-03 | 0.1080 | 9.1e-12 | 0.36 |
| September | 1.1e-03 | 0.1434 | 0.48 | 2.1e-03 | 0.4980 | 2.1e+00 | 0.53 |
| All | 3.3e-03 | 0.1229 | 0.37 | 3.3e-03 | 0.1229 | 9.7e-25 | 0.39 |
| **Birch** | | | | | | | |
| May | 3.8e-03 | 0.0228 | 0.33 | 2.6e-03 | 0.0032 | 4.2e-01 | 0.35 |
| June | 2.2e-03 | 0.0151 | 0.16 | 1.6e-03 | 0.0000 | 2.5e-01 | 0.28 |
| July | 1.4e-03 | 0.0438 | 0.71 | 1.4e-03 | 0.0438 | 2.0e-06 | 0.69 |
| August | 4.7e-03 | 0.0217 | 0.01 | 2.9e-04 | 0.0000 | 8.0e-01 | 0.08 |
| All | 2.2e-03 | 0.0224 | 0.23 | 1.4e-03 | 0.0000 | 3.3e-01 | 0.31 |





## 4    Conclusions

This study presents the Total OH Reactivity of Emissions (TOHRE) for three tree species from the boreal forest. The studied trees were seedlings (in pots) placed outside the measurement container at the SMEAR II station in Hyytiälä, Finland. Instruments to measure TOHRE with the comparative reactivity method (CRM) and the chemical composition of the emissions (two on-line GC-MS systems) were located inside the container. Three dynamic branch enclosure (one for each tree species) were set up, but VOC and TOHRE measurements were performed from one enclosure at a time for periods ranging from a few days
to over a week.

The results show that the chemical composition of the emissions varies greatly between tree species but also for the same tree depending on environmental conditions. The emissions of the seedlings were classified as stress-induced on several occasions. During these periods, TOHRE increased greatly and did not return to values close to zero at night and the emission profiles changed with an increased fraction of Green Leaf Volatiles (GLVs) and different terpene emissions.

Pine emissions were dominated by monoterpenes for all measurement periods with varying fractions of MBO and sesquiterpenes mostly. GLVs were found to be up to almost 40 % of the known reactivity in July and October for two short stress periods. Spruce emissions were also dominated by monoterpenes and from July onwards sesquiterpenes contributed almost equally to TOHRE. Exceptions are the two stress periods, where GLVs and aldehydes were the major compounds. Birch emissions were dominated by various fractions of monoterpenes and sesquiterpenes with GLVs also present, especially in mid-July
and August.

In absolute terms the highest TOHRE values were measured for birch, mostly due to the low reactivity (because of the small biomass) and higher influence of the measurement background, compared to the other two tree species. Also higher TOHRE averages were found for spruce, compared to pine, indicating that knowledge of the tree composition of a forest is important in order to assess reactive emissions.

In general the missing OHRE fraction remain high, but for pine and spruce it was driven by low reactivity periods (low COHRE and scatter of the TOHRE measurements) and the missing OHRE fraction was smaller for periods with higher TOHRE. However for birch, we found consistently high missing fraction throughout the measurement periods, which emphasise the need to look for emitted compounds with different functionalities than the ones studied so far.

Moreover, TOHRE exhibited various degrees of temperature dependence. In particular for spruce this temperature depen-
dence had a strong seasonality: a high temperature dependence was found in July and August (when less volatile compounds are emitted, such as sesquiterpenes), but a low dependence was measured in May, and September. For pine and birch the temperature difference was varying less with the seasons. Stress emissions for pine in July were not temperature dependent at all and no correlation could be found. Taking into account photosynthetically active radiation (PAR) with an hybrid model did not improve significantly the correlations, except for the notable exception of pine emissions in July (including very large peak on
9 July).

Because this type of characterization of TOHRE is rare, only a comparison with a study by Nölscher et al. (2013) is possible. They found that the missing OHRE fraction for spruce emissions was low in spring and increased as the seasons proceeded,





while in the present study we found a larger missing OHRE fraction for spruce emissions in the spring compared to the later time of the year. This underscores how much is still unknown regarding biogenic emissions of reactive species, but also the challenges of the methods used. For instance, Nölscher et al. (2013) did not have continuous GC-MS measurements throughout the year and relied on a constant chemical speciation derived in the spring, while our results demonstrate that emission profiles vary throughout the year and react to various environmental conditions, in particular stress episodes. Further understanding, characterization and quantification of such stress episodes (and their many causes) is necessary in order to better model reactive emissions from vegetation in global model as they can occur suddenly and with high intensity.

While it remains difficult to generalise from the particular dataset presented in this study, clear future research directions are highlighted. In addition, direct in situ studies for various trees from the forest should be conducted to confirm the findings of the present work.





## Appendix A: Pictures of the branches

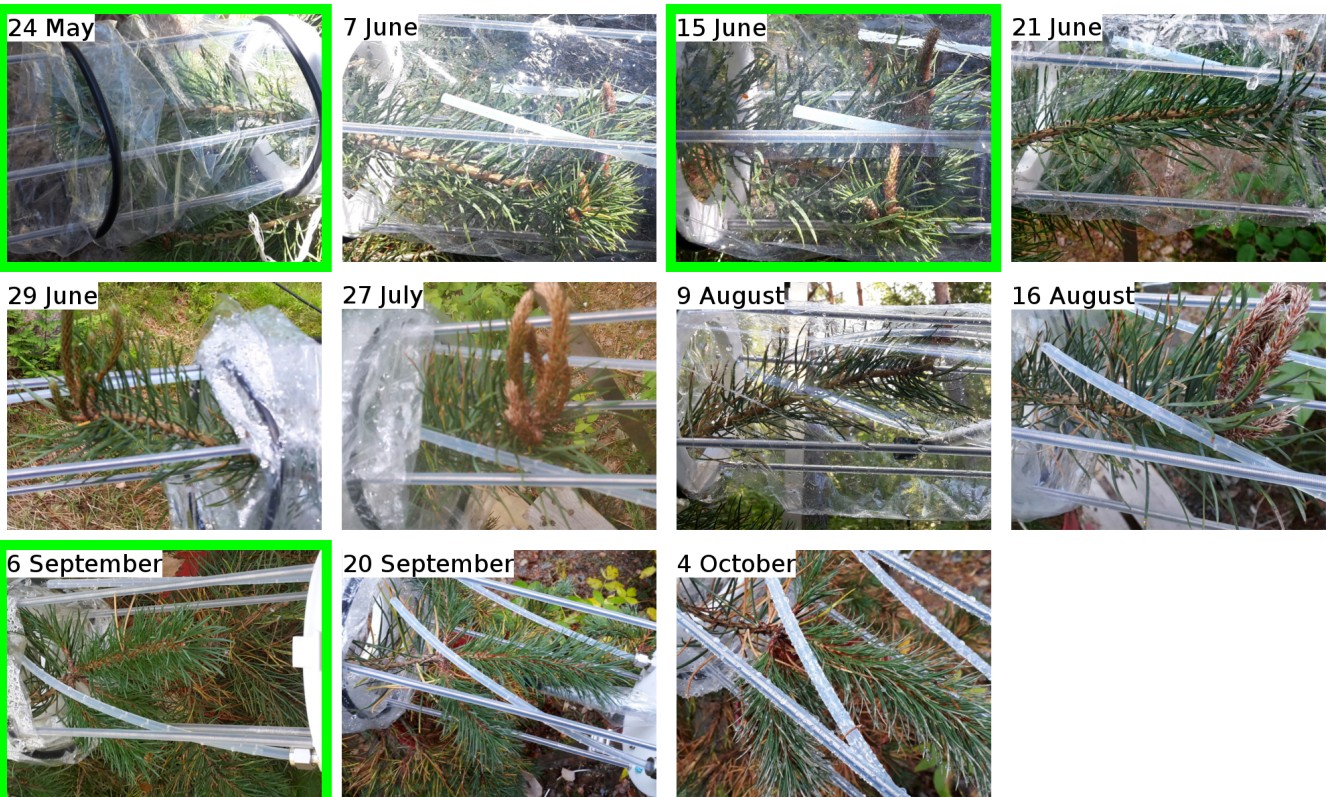

**Figure A1.** Pictures of pine branches. Dates framed in green indicate a new branch placed in the enclosure.



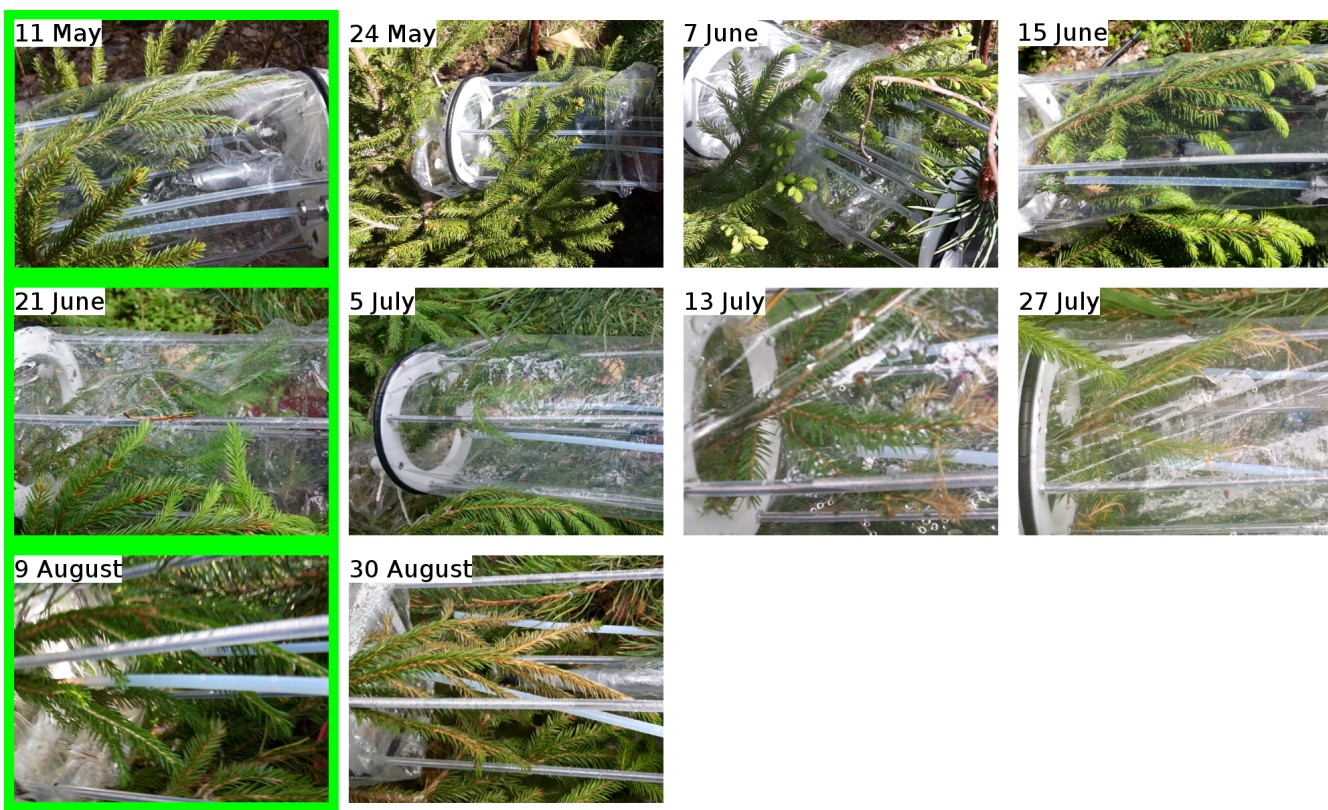

**Figure A2.** Pictures of spruce branches. Dates framed in green indicate a new branch placed in the enclosure.





**Figure A3.** Pictures of birch branches. Dates framed in green indicate a new branch placed in the enclosure.





## Appendix B: Dry weight of biomass

**Table B1.** Dry weight of the needles' or leaves' biomass on the dates the branches were cut.

| | Pine | | Spruce | | Birch | |
|---|---|---|---|---|---|---|
| 15 June | 9.2 g + 0.8g (buds) | | 21 June | 7.62 g | 7 June | 0.5454 g |
| 16 August | 5.94 g + 1.3g (buds) | | 9 August | 2.3 g | 9 August | 1.32 g |
| 11 October | 5.133 g | | 5 November | 5.616 | 6 September | - |

## Appendix C: Dynamic branch enclosure

### C1 Temperature difference inside the enclosure compare to ambient

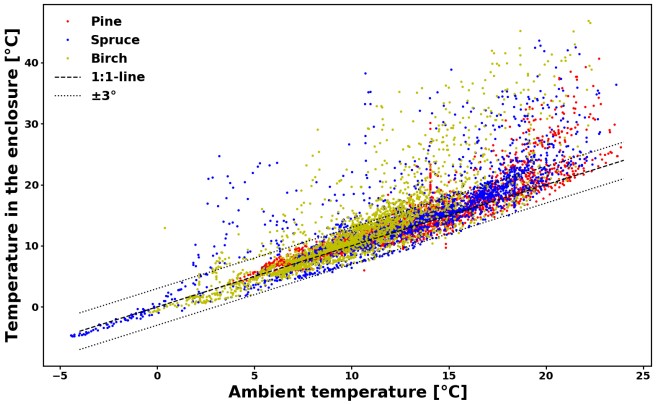

**Figure C1.** Temperature inside the encolusre compared against ambient temperature.

### C2 Blank reactivity values





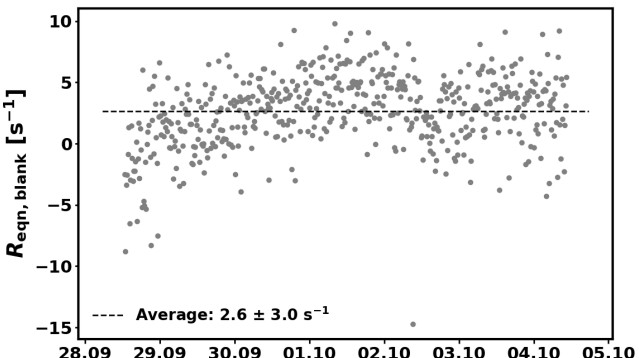

**Figure C2.** $R_{\mathrm{eqn,blank}}$ measured from an empty branch enclosure.



## Appendix D: COHRE by compound

Table D1: Averages of individual compounds OH reactivity of the emissions, OHRE [$m^3\,s^{-2}\,g_{dw}^{-1}$], with standard deviations (in brackets) for the different measurement periods for pine. 'n.d.' means 'not detected'.

| | Period P1 7–14 June | Period P2.a 29 June–5 July | Period P2.b 19–21 July | Period P2.c 10–16 August | Period P3.a 13–20 September | Period P3.b 4–11 October |
|---|---|---|---|---|---|---|
| isoprene | 1.5E-07 (± 1.6E-07) | 4.9E-07 (± 8.3E-07) | 3.1E-07 (± 4.0E-07) | 2.1E-07 (± 3.6E-07) | 8.7E-08 (± 5.5E-08) | 4.5E-08 (± 2.7E-08) |
| MBO | 9.8E-07 (± 1.1E-06) | 6.4E-06 (± 1.0E-05) | 2.5E-06 (± 2.9E-06) | 1.2E-06 (± 2.2E-06) | 4.6E-07 (± 4.7E-07) | 4.4E-07 (± 3.2E-07) |
| $\alpha$-pinene | 4.0E-06 (± 4.2E-06) | 2.6E-05 (± 2.5E-05) | 1.2E-05 (± 7.2E-06) | 7.9E-06 (± 6.7E-06) | 6.1E-06 (± 3.2E-06) | 1.1E-05 (± 1.0E-05) |
| $\beta$-pinene | 5.8E-07 (± 1.4E-06) | 3.7E-06 (± 2.7E-06) | 4.4E-06 (± 2.7E-06) | 2.7E-06 (± 2.5E-06) | 1.4E-06 (± 7.4E-07) | 2.2E-06 (± 1.8E-06) |
| camphene | 5.2E-07 (± 5.5E-07) | 4.8E-06 (± 3.4E-06) | 3.3E-06 (± 1.9E-06) | 1.7E-06 (± 1.5E-06) | 1.8E-06 (± 9.2E-07) | 3.0E-06 (± 2.3E-06) |
| $\Delta^3$-carene | 2.1E-06 (± 4.0E-06) | 2.1E-05 (± 2.4E-05) | 1.0E-05 (± 6.0E-06) | 6.9E-06 (± 6.6E-06) | 6.0E-06 (± 3.2E-06) | 1.1E-05 (± 9.1E-06) |
| $\beta$-phellandrene[a] | 3.0E-07 (± 1.1E-06) | 1.7E-06 (± 1.8E-06) | 3.9E-06 (± 2.6E-06) | 1.7E-06 (± 2.0E-06) | 3.5E-07 (± 2.1E-07) | 6.0E-07 (± 5.6E-07) |
| $p$-cymene | 1.5E-08 (± 3.0E-08) | 1.0E-07 (± 1.0E-07) | 9.4E-08 (± 6.0E-08) | 3.9E-08 (± 4.3E-08) | 4.7E-08 (± 2.7E-08) | 7.8E-08 (± 5.1E-08) |
| 1,8-cineol | 1.0E-09 (± 4.1E-09) | 3.5E-08 (± 4.7E-08) | 5.2E-09 (± 3.5E-08) | 7.8E-09 (± 7.9E-09) | 2.4E-08 (± 1.6E-08) | 3.7E-08 (± 1.8E-08) |
| limonene | 2.1E-06 (± 5.5E-06) | 6.6E-06 (± 6.6E-06) | 9.9E-06 (± 6.1E-06) | 6.3E-06 (± 7.0E-06) | 2.2E-06 (± 1.2E-06) | 2.8E-06 (± 2.8E-06) |
| terpinolene | 9.2E-07 (± 1.1E-06) | 7.7E-06 (± 9.7E-06) | 4.6E-06 (± 3.1E-06) | 1.1E-06 (± 1.1E-06) | 2.0E-06 (± 1.2E-06) | 4.6E-06 (± 4.3E-06) |
| borneol[b] | 3.0E-09 (± 1.6E-08) | n.d. | n.d. | n.d. | n.d. | n.d. |
| linalool | 9.2E-10 (± 1.2E-08) | 3.8E-08 (± 8.2E-08) | n.d. | 1.2E-08 (± 2.5E-08) | 2.5E-08 (± 2.3E-08) | 1.1E-07 (± 5.6E-08) |
| myrcene | 4.8E-16 (± 6.5E-16) | n.d. | 8.1E-15 (± 4.9E-15) | 2.3E-15 (± 2.4E-15) | n.d. | 4.9E-15 (± 4.8E-15) |
| $\alpha$-terpineol[c] | 2.2E-09 (± 1.5E-08) | 3.0E-08 (± 1.3E-07) | n.d. | n.d. | 1.0E-08 (± 5.1E-08) | 2.0E-07 (± 2.3E-07) |
| longicyclene | 1.1E-11 (± 2.8E-10) | 2.2E-10 (± 1.2E-09) | 1.0E-09 (± 8.6E-09) | n.d. | 8.8E-11 (± 4.1E-10) | 6.8E-10 (± 1.6E-09) |
| isolongifolene and agurjunene | 1.7E-10 (± 3.0E-09) | 3.6E-09 (± 1.2E-08) | 8.9E-09 (± 7.4E-08) | 9.1E-11 (± 8.0E-10) | 5.3E-10 (± 3.1E-09) | 2.7E-08 (± 1.6E-08) |
| $\beta$-bourbonene[d] | 9.7E-10 (± 5.5E-09) | n.d. | n.d. | n.d. | n.d. | n.d. |
| $\beta$-farnesene | 6.4E-08 (± 1.5E-07) | 2.4E-06 (± 3.0E-06) | n.d. | 1.6E-08 (± 3.5E-08) | 3.1E-07 (± 2.2E-07) | 2.6E-07 (± 2.3E-07) |
| $\alpha$-farnesene[e] | 3.0E-08 (± 1.4E-07) | 1.2E-06 (± 1.1E-06) | n.d. | 2.7E-07 (± 2.9E-07) | 5.5E-06 (± 3.3E-06) | 4.4E-06 (± 2.0E-06) |
| $\beta$-caryophyllene | 1.3E-07 (± 4.7E-07) | 8.3E-07 (± 1.2E-06) | n.d. | 6.2E-08 (± 8.4E-08) | 2.3E-08 (± 5.7E-08) | 1.1E-06 (± 8.0E-07) |
| $\alpha/\beta$-cubebene[f] | 9.8E-10 (± 8.8E-09) | n.d. | n.d. | n.d. | n.d. | n.d. |
| $\alpha$-humulene | 9.9E-09 (± 4.1E-08) | 9.7E-08 (± 1.7E-07) | n.d. | 3.3E-09 (± 9.7E-09) | 5.1E-08 (± 4.4E-07) | 1.9E-07 (± 1.5E-07) |
| SQT1[f] | 6.1E-08 (± 2.8E-07) | 2.4E-08 (± 8.1E-08) | n.d. | 2.8E-10 (± 1.7E-09) | 4.8E-09 (± 2.3E-08) | 6.8E-08 (± 8.5E-08) |
| SQT2[f] | 2.2E-07 (± 9.2E-07) | 1.6E-06 (± 1.1E-06) | n.d. | 9.9E-09 (± 2.4E-08) | 9.6E-08 (± 6.2E-08) | n.d. |
| SQT3[f] | 1.4E-07 (± 3.9E-07) | 6.3E-07 (± 1.6E-06) | n.d. | 3.0E-07 (± 3.9E-07) | 6.7E-08 (± 4.1E-08) | 4.6E-07 (± 4.0E-07) |
| SQT4[g] | 2.1E-07 (± 4.5E-07) | 1.7E-06 (± 1.7E-06) | n.d. | 7.0E-07 (± 8.1E-07) | 1.8E-07 (± 9.1E-08) | 9.4E-07 (± 7.7E-07) |
| SQT5[h] | n.d. | 8.8E-09 (± 5.2E-08) | n.d. | n.d. | 3.6E-08 (± 1.0E-07) | n.d. |
| SQT6[f] | 1.8E-10 (± 3.3E-09) | 3.3E-09 (± 1.9E-08) | n.d. | n.d. | 1.2E-08 (± 2.0E-08) | n.d. |
| SQT7[f] | n.d. | 2.5E-08 (± 9.9E-08) | n.d. | 8.3E-09 (± 1.7E-08) | 3.2E-09 (± 9.7E-09) | 1.4E-07 (± 1.1E-07) |
| SQT8[f] | 1.5E-10 (± 2.7E-09) | 3.5E-08 (± 4.2E-08) | n.d. | n.d. | n.d. | n.d. |
| SQT9[f] | 2.1E-09 (± 1.1E-08) | 3.7E-07 (± 5.7E-07) | n.d. | 3.0E-08 (± 4.3E-08) | n.d. | n.d. |
| SQT10[g] | 1.6E-08 (± 6.4E-08) | 1.2E-07 (± 2.4E-07) | n.d. | 4.0E-08 (± 4.7E-08) | n.d. | n.d. |
| SQT11[d] | 5.3E-08 (± 1.9E-07) | n.d. | n.d. | n.d. | n.d. | n.d. |
| SQT12[f] | 2.1E-08 (± 7.6E-08) | n.d. | n.d. | n.d. | n.d. | n.d. |
| SQT13[d] | 2.1E-08 (± 7.9E-08) | n.d. | n.d. | n.d. | n.d. | n.d. |
| SQT14[d] | 1.2E-08 (± 5.4E-08) | n.d. | n.d. | n.d. | n.d. | n.d. |
| SQT15[d] | 1.0E-09 (± 2.5E-08) | n.d. | n.d. | n.d. | n.d. | n.d. |
| 1-hexanol | 2.4E-10 (± 4.2E-09) | 1.5E-07 (± 3.3E-07) | n.d. | n.d. | 1.8E-09 (± 1.2E-08) | 8.5E-08 (± 1.6E-07) |
| $cis$-2-hexen-1-ol | n.d. | n.d. | n.d. | n.d. | n.d. | n.d. |
| $trans$-2-hexen-1-ol | n.d. | n.d. | n.d. | n.d. | n.d. | n.d. |





|  | Period P1 7–14 June | Period P2.a 29 June–5 July | Period P2.b 19–21 July | Period P2.c 10–16 August | Period P3.a 13–20 September | Period P3.b 4–11 October |
|---|---|---|---|---|---|---|
| *cis*-3-hexen-1-ol | n.d. | 5.6E-06 (± 1.3E-05 ) | n.d. | n.d. | 5.8E-10 (± 1.0E-08 ) | 3.5E-06 (± 5.6E-06 ) |
| *trans*-3-hexen-1-ol | n.d. | n.d. | n.d. | n.d. | 4.0E-09 (± 5.0E-08 ) | n.d. |
| hexyl acetate | n.d. | n.d. | n.d. | n.d. | n.d. | n.d. |
| *cis*-3-hexenyl acetate | n.d. | 1.7E-07 (± 1.1E-06 ) | n.d. | n.d. | 9.4E-09 (± 5.2E-08 ) | n.d. |
| *trans*-2-hexenyl acetate | n.d. | n.d. | n.d. | n.d. | n.d. | n.d. |
| pentanal | 2.1E-08 (± 1.2E-08 ) | 4.6E-08 (± 5.3E-08 ) | n.d. | 4.3E-08 (± 3.7E-08 ) | 7.5E-09 (± 4.5E-09 ) | 9.9E-08 (± 6.1E-08 ) |
| hexanal | 1.5E-08 (± 1.1E-08 ) | 1.5E-06 (± 3.0E-06 ) | 2.8E-07 (± 1.9E-07 ) | 8.4E-08 (± 9.8E-08 ) | 1.2E-08 (± 8.2E-09 ) | 5.0E-07 (± 5.6E-07 ) |
| heptanal | 1.3E-08 (± 1.3E-08 ) | 2.6E-08 (± 3.2E-08 ) | n.d. | 2.8E-08 (± 2.8E-08 ) | 2.1E-08 (± 1.4E-08 ) | 2.0E-07 (± 1.0E-07 ) |
| octanal | 2.3E-08 (± 2.4E-08 ) | 3.7E-08 (± 5.0E-08 ) | n.d. | 2.2E-08 (± 2.9E-08 ) | 9.2E-09 (± 7.3E-09 ) | 7.3E-08 (± 4.2E-08 ) |
| nonanal | 4.2E-08 (± 3.4E-08 ) | 9.3E-08 (± 9.6E-08 ) | n.d. | 6.7E-08 (± 6.7E-08 ) | 2.1E-08 (± 1.6E-08 ) | 1.1E-07 (± 6.0E-08 ) |
| decanal | 5.5E-08 (± 3.2E-08 ) | 4.9E-08 (± 5.5E-08 ) | n.d. | 2.3E-08 (± 3.0E-08 ) | 1.4E-08 (± 1.1E-08 ) | 4.9E-08 (± 2.6E-08 ) |
| methacrolein | 2.2E-09 (± 7.3E-09 ) | 1.5E-08 (± 2.1E-08 ) | n.d. | 2.2E-08 (± 2.1E-08 ) | 2.6E-09 (± 3.1E-09 ) | 4.3E-08 (± 2.7E-08 ) |
| 1-pentanol | 2.2E-10 (± 3.9E-09 ) | 1.4E-09 (± 2.3E-08 ) | n.d. | n.d. | 1.1E-10 (± 1.9E-09 ) | 6.6E-09 (± 4.3E-08 ) |
| 1-octen-3-ol | n.d. | n.d. | n.d. | n.d. | n.d. | n.d. |
| butyl acetate | n.d. | n.d. | n.d. | n.d. | n.d. | n.d. |
| bornyl acetate | 5.2E-08 (± 1.5E-07 ) | 9.7E-08 (± 2.2E-07 ) | 3.7E-09 (± 2.1E-08 ) | 1.4E-08 (± 1.5E-08 ) | 2.7E-08 (± 1.4E-08 ) | 9.1E-08 (± 1.2E-07 ) |
| propanoic acid | 6.5E-09 (± 1.3E-08 ) | 2.3E-10 (± 2.2E-09 ) | n.d. | n.d. | 7.1E-09 (± 8.3E-09 ) | 6.7E-10 (± 5.1E-09 ) |
| butanoic acid | 2.4E-08 (± 1.6E-08 ) | 7.0E-09 (± 8.6E-09 ) | n.d. | n.d. | 1.2E-08 (± 7.4E-09 ) | 1.2E-08 (± 1.5E-08 ) |
| isobutanoic acid | 2.2E-09 (± 9.9E-09 ) | 1.9E-09 (± 1.1E-08 ) | n.d. | n.d. | 1.1E-08 (± 1.3E-08 ) | 1.0E-08 (± 3.5E-08 ) |
| pentanoic acid | n.d. | n.d. | n.d. | n.d. | 1.4E-09 (± 6.3E-09 ) | n.d. |
| isopentanoic acid | n.d. | n.d. | n.d. | n.d. | 3.7E-10 (± 2.2E-09 ) | n.d. |
| hexanoic acid | n.d. | n.d. | n.d. | n.d. | 2.2E-10 (± 3.8E-09 ) | n.d. |
| 4-methylpentanoic acid | n.d. | n.d. | n.d. | n.d. | n.d. | n.d. |
| heptanoicacid | n.d. | n.d. | n.d. | n.d. | n.d. | n.d. |

[a] quantified as $\Delta^3$-carene    [b] quantified as bornylacetate    [c] quantified as terpinolene    [d] quantified as isolongifolene    [e] quantified as $\beta$-farnesene    [f] quantified as $\beta$-caryophyllene    [g] quantified as $\beta$-caryophyllene or isolongifolene    [h] quantified as longicyclene

Table D2: Averages of individual compounds OH reactivity of the emissions, OHRE [$m^3\,s^{-2}\,g_{dw}^{-1}$], with standard deviations (in brackets) for the different measurement periods for spruce. 'n.d.' means 'not detected' and 'n.m.' means 'not measured'.

|  | Period S1.a 11–14 May | Period S1.b 19–24 May | Period S2 5–13 July | Period S3.a 16–23 August | Period S3.b 6–13 September |
|---|---|---|---|---|---|
| isoprene | 1.3E-07 (± 1.9E-07 ) | 1.0E-06 (± 2.3E-06 ) | 7.3E-07 (± 1.0E-06 ) | 8.9E-08 (± 1.2E-07 ) | 6.4E-08 (± 7.2E-08 ) |
| MBO | 1.8E-07 (± 2.8E-07 ) | 6.5E-07 (± 1.2E-06 ) | 8.3E-07 (± 1.3E-06 ) | 2.1E-07 (± 2.8E-07 ) | 1.1E-07 (± 1.3E-07 ) |
| $\alpha$-pinene | 6.9E-06 (± 1.1E-05 ) | 9.7E-07 (± 2.0E-06 ) | 1.8E-05 (± 3.8E-05 ) | 3.8E-07 (± 3.9E-07 ) | 1.1E-06 (± 9.7E-07 ) |
| $\beta$-pinene | 1.4E-05 (± 2.2E-05 ) | 8.9E-07 (± 2.5E-06 ) | 1.4E-05 (± 1.9E-05 ) | 2.5E-07 (± 2.2E-07 ) | 1.3E-06 (± 1.2E-06 ) |
| camphene | 3.7E-06 (± 6.0E-06 ) | 3.4E-07 (± 6.5E-07 ) | 2.7E-05 (± 6.0E-05 ) | 4.8E-07 (± 7.7E-07 ) | 1.2E-06 (± 9.8E-07 ) |
| $\Delta^3$-carene | 3.8E-06 (± 8.6E-06 ) | 1.7E-07 (± 4.4E-07 ) | 5.9E-06 (± 7.3E-06 ) | 6.9E-08 (± 3.9E-08 ) | 2.2E-07 (± 2.2E-07 ) |
| $\beta$-phellandrene[a] | 1.5E-05 (± 2.3E-05 ) | 7.0E-07 (± 1.9E-06 ) | 8.0E-06 (± 1.4E-05 ) | 9.2E-08 (± 5.5E-08 ) | 9.8E-07 (± 1.1E-06 ) |
| *p*-cymene | 2.2E-07 (± 4.7E-07 ) | 1.5E-08 (± 3.1E-08 ) | 3.8E-07 (± 7.7E-07 ) | 3.8E-09 (± 2.5E-09 ) | 7.2E-09 (± 5.7E-09 ) |
| 1,8-cineol | 6.0E-07 (± 7.2E-07 ) | 6.0E-08 (± 1.1E-07 ) | 4.9E-06 (± 1.2E-05 ) | 3.4E-08 (± 2.6E-08 ) | 2.0E-07 (± 1.8E-07 ) |
| limonene | 3.4E-05 (± 4.4E-05 ) | 3.6E-06 (± 6.2E-06 ) | 1.1E-04 (± 2.5E-04 ) | 2.9E-06 (± 2.4E-06 ) | 1.7E-05 (± 1.3E-05 ) |
| terpinolene | 4.1E-06 (± 8.1E-06 ) | 2.7E-07 (± 5.8E-07 ) | 6.2E-06 (± 1.8E-05 ) | 6.1E-08 (± 4.2E-08 ) | 1.8E-07 (± 2.1E-07 ) |



| | Period S1.a 11–14 May | Period S1.b 19–24 May | Period S2 5–13 July | Period S3.a 16–23 August | Period S3.b 6–13 September |
|---|---|---|---|---|---|
| borneol[b] | n.d. | n.d. | n.d. | n.d. | n.d. |
| linalool | 3.4E-08 (± 1.2E-07) | 6.2E-09 (± 2.3E-08) | 1.1E-06 (± 4.5E-06) | 3.0E-08 (± 3.0E-08) | 2.2E-08 (± 1.7E-08) |
| myrcene | 2.9E-14 (± 6.3E-14) | 9.5E-16 (± 2.2E-15) | n.d. | 6.6E-16 (± 5.0E-16) | n.d. |
| $\alpha$-terpineol[c] | n.d. | n.d. | 3.2E-06 (± 1.1E-05) | 2.2E-08 (± 2.6E-08) | 1.4E-07 (± 1.1E-07) |
| longicyclene | 2.7E-09 (± 5.6E-09) | 3.0E-10 (± 1.1E-09) | 2.0E-10 (± 2.4E-09) | 4.7E-10 (± 4.6E-10) | 6.2E-10 (± 1.6E-09) |
| isolongifolene and agurjunene | 1.1E-09 (± 4.2E-09) | 6.9E-09 (± 2.6E-08) | 6.8E-09 (± 3.1E-08) | 1.7E-09 (± 3.2E-09) | 3.6E-10 (± 1.7E-09) |
| $\beta$-farnesene | 1.9E-07 (± 2.5E-07) | 2.0E-07 (± 2.5E-07) | 9.1E-06 (± 1.1E-05) | 3.8E-06 (± 4.1E-06) | 2.1E-06 (± 1.4E-06) |
| $\alpha$-farnesene[d] | 2.1E-07 (± 5.4E-07) | 9.0E-07 (± 1.2E-06) | 4.7E-05 (± 6.2E-05) | 1.9E-05 (± 2.3E-05) | 1.4E-05 (± 6.8E-06) |
| $\alpha$-humulene | 4.1E-07 (± 6.6E-07) | 9.5E-08 (± 2.2E-07) | 1.7E-07 (± 4.9E-07) | 2.5E-08 (± 3.6E-08) | 6.8E-08 (± 4.5E-08) |
| $\beta$-caryophyllene | 1.0E-06 (± 2.0E-06) | 2.1E-07 (± 5.7E-07) | 2.6E-07 (± 6.5E-07) | 6.6E-08 (± 9.7E-08) | 1.2E-07 (± 6.9E-08) |
| SQT1[e] | 1.5E-08 (± 3.6E-08) | n.d. | 8.2E-08 (± 1.5E-07) | 1.3E-09 (± 6.5E-09) | 1.3E-08 (± 4.5E-08) |
| SQT2[e] | 6.6E-08 (± 1.8E-07) | 2.4E-07 (± 3.4E-07) | 1.0E-06 (± 1.7E-06) | 1.2E-10 (± 2.1E-09) | 1.0E-07 (± 4.8E-08) |
| SQT3[e] | n.d. | n.d. | 2.7E-07 (± 4.3E-07) | 2.3E-08 (± 4.1E-08) | 1.1E-08 (± 1.9E-08) |
| SQT4[f] | 2.3E-07 (± 3.6E-07) | 9.1E-08 (± 2.4E-07) | 3.3E-07 (± 7.1E-07) | 2.1E-08 (± 1.8E-08) | 2.2E-08 (± 1.2E-08) |
| SQT5[g] | 1.7E-07 (± 4.4E-07) | n.d. | 2.0E-07 (± 6.7E-07) | 4.9E-08 (± 7.2E-08) | 3.5E-08 (± 7.5E-08) |
| SQT6[e] | 4.2E-07 (± 7.4E-07) | 3.9E-08 (± 1.3E-07) | 8.3E-08 (± 1.3E-07) | 5.4E-11 (± 9.5E-10) | 4.9E-08 (± 3.2E-08) |
| SQT7[e] | n.d. | n.d. | 7.1E-08 (± 1.4E-07) | 6.4E-09 (± 1.1E-08) | 7.5E-09 (± 1.3E-08) |
| SQT8[e] | n.d. | n.d. | n.d. | n.d. | 1.9E-09 (± 7.2E-09) |
| SQT9[e] | n.d. | n.d. | n.d. | n.d. | n.d. |
| SQT10[f] | n.d. | n.d. | n.d. | n.d. | 7.2E-11 (± 1.3E-09) |
| 1-hexanol | n.m. | n.m. | 1.3E-06 (± 3.5E-06) | 1.1E-06 (± 1.3E-06) | 1.8E-08 (± 2.6E-08) |
| cis-2-hexen-1-ol | n.m. | n.m. | n.d. | n.d. | n.d. |
| trans-2-hexen-1-ol | n.m. | n.m. | 1.0E-06 (± 4.1E-06) | n.d. | n.d. |
| cis-3-hexen-1-ol | n.m. | n.m. | 3.2E-04 (± 9.9E-04) | 8.0E-05 (± 8.1E-05) | 8.1E-07 (± 6.0E-07) |
| trans-3-hexen-1-ol | n.m. | n.m. | 3.4E-07 (± 2.9E-06) | 9.1E-08 (± 3.8E-07) | 1.5E-08 (± 9.2E-08) |
| trans-2-hexenal | n.d. | n.d. | 3.4E-05 (± 4.8E-05) | 3.5E-05 (± 4.5E-05) | 1.8E-07 (± 1.4E-07) |
| hexyl acetate | n.m. | n.m. | 3.3E-08 (± 1.5E-07) | 2.0E-07 (± 2.7E-07) | n.d. |
| cis-3-hexenyl acetate | n.m. | n.m. | 8.4E-05 (± 2.3E-04) | 6.2E-05 (± 6.9E-05) | 2.9E-07 (± 2.6E-07) |
| trans-2-hexenyl acetate | n.m. | n.m. | n.d. | n.d. | n.d. |
| pentanal | 2.8E-08 (± 2.4E-08) | 1.4E-08 (± 1.9E-08) | 8.8E-06 (± 5.0E-06) | 2.5E-07 (± 3.4E-07) | 1.5E-08 (± 1.1E-08) |
| hexanal | 5.9E-08 (± 7.4E-08) | 2.5E-08 (± 4.3E-08) | 2.3E-06 (± 3.5E-06) | 1.0E-05 (± 1.5E-05) | 3.1E-08 (± 2.2E-08) |
| heptanal | 3.4E-08 (± 1.8E-08) | 1.8E-08 (± 2.1E-08) | 8.8E-06 (± 4.4E-06) | 9.1E-08 (± 5.2E-08) | 5.1E-08 (± 5.1E-08) |
| octanal | 4.6E-08 (± 5.3E-08) | 1.3E-08 (± 1.6E-08) | 8.4E-06 (± 4.8E-06) | 1.2E-07 (± 9.6E-08) | 2.4E-08 (± 2.6E-08) |
| nonanal | 1.2E-07 (± 1.7E-07) | 2.5E-08 (± 3.0E-08) | 1.0E-05 (± 5.0E-06) | 6.9E-08 (± 4.4E-08) | 2.6E-08 (± 2.5E-08) |
| decanal | 1.4E-07 (± 1.3E-07) | 4.0E-08 (± 4.3E-08) | 5.1E-06 (± 2.9E-06) | 5.5E-08 (± 3.3E-08) | 1.3E-08 (± 1.3E-08) |
| methacrolein | 1.1E-08 (± 1.4E-08) | 5.7E-09 (± 1.3E-08) | 9.7E-06 (± 4.3E-06) | 6.8E-08 (± 9.5E-08) | 1.2E-08 (± 7.4E-09) |
| 1-pentanol | n.m. | n.m. | 2.3E-07 (± 1.1E-06) | 6.4E-08 (± 1.2E-07) | n.d. |
| 1-octen-3-ol | n.m. | n.m. | 5.2E-08 (± 3.0E-07) | n.d. | n.d. |
| butyl acetate | n.m. | n.m. | n.d. | n.d. | n.d. |
| bornyl acetate | 1.4E-07 (± 1.8E-07) | 3.8E-08 (± 7.3E-08) | 3.9E-06 (± 1.0E-05) | 4.9E-08 (± 6.1E-08) | 8.1E-08 (± 5.3E-08) |
| propanoic acid | n.m. | n.m. | 1.4E-08 (± 6.0E-08) | 1.4E-08 (± 2.3E-08) | 7.6E-09 (± 6.7E-09) |
| butanoic acid | n.m. | n.m. | 1.2E-07 (± 1.1E-07) | 4.6E-08 (± 3.2E-08) | 1.1E-08 (± 6.7E-09) |
| isobutanoic acid | n.m. | n.m. | 2.0E-08 (± 8.2E-08) | 4.3E-08 (± 4.2E-08) | 6.8E-09 (± 9.9E-09) |
| pentanoic acid | n.m. | n.m. | 3.4E-10 (± 8.9E-09) | 2.2E-09 (± 1.3E-08) | 2.0E-10 (± 2.1E-09) |
| isopentanoic acid | n.m. | n.m. | n.d. | n.d. | n.d. |
| hexanoic acid | n.m. | n.m. | n.d. | n.d. | n.d. |



|  | Period S1.a<br>11–14 May | Period S1.b<br>19–24 May | Period S2<br>5–13 July | Period S3.a<br>16–23 August | Period S3.b<br>6–13 September |
|---|---|---|---|---|---|
| 4-methylpentanoic acid | n.m. | n.m. | n.d. | n.d. | n.d. |

[a] quantified as carene  [b] quantified as bornylacetate  [c] quantified as terpinolene  [d] quantified as $\beta$-farnesene  [e] quantified as $\beta$-caryophyllene  [f] quantified as $\beta$-caryophyllene or isolongifolene  [g] quantified as longicyclene

Table D3: Averages of individual compounds OH reactivity of the emissions, OHRE [$m^3\,s^{-2}\,g_{dw}^{-1}$], with standard deviations (in brackets) for the different measurement periods for birch. 'n.d.' means 'not detected'.

|  | Period B1<br>24 May–8 June | Period B2.a<br>21–29 June | Period B2.b<br>13–19 July | Period B3<br>23–28 August |
|---|---|---|---|---|
| isoprene | 8.1E-09 (± 2.2E-08 ) | 1.5E-09 (± 1.1E-08 ) | 5.25E-08 (± 1.88E-07 ) | 1.19E-08 (± 3.05E-08 ) |
| MBO | 2.5E-08 (± 7.0E-08 ) | 1.2E-09 (± 2.0E-08 ) | 1.22E-08 (± 8.55E-08 ) | 2.66E-09 (± 1.78E-08 ) |
| $\alpha$-pinene | 3.5E-07 (± 7.2E-07 ) | 9.20E-08 (± 7.01E-08 ) | 1.13E-07 (± 1.18E-07 ) | 2.08E-06 (± 3.62E-06 ) |
| $\beta$-pinene | 1.0E-08 (± 2.0E-08 ) | 5.6E-09 (± 4.0E-08 ) | 4.57E-10 (± 7.45E-09 ) | 1.63E-07 (± 2.75E-07 ) |
| camphene | 7.9E-09 (± 1.7E-08 ) | 2.5E-09 (± 1.8E-08 ) | 1.18E-09 (± 1.41E-08 ) | 5.42E-08 (± 8.29E-08 ) |
| carene | 7.5E-08 (± 1.6E-07 ) | 5.26E-09 (± 3.34E-08 ) | 1.01E-08 (± 8.00E-08 ) | 2.54E-06 (± 3.96E-06 ) |
| $\beta$-phellandrene[a] | 1.7E-08 (± 3.2E-08 ) | n.d. | n.d. | 9.09E-08 (± 1.52E-07 ) |
| $p$-cymene | 4.3E-09 (± 1.0E-08 ) | 7.0E-11 (± 1.2E-09 ) | n.d. | 6.99E-09 (± 1.29E-08 ) |
| 1,8-cineol | 3.6E-09 (± 9.0E-09 ) | n.d. | 1.13E-08 (± 5.41E-08 ) | 5.27E-10 (± 3.68E-09 ) |
| limonene | 4.0E-07 (± 1.0E-06 ) | 5.2E-07 (± 8.6E-07 ) | 8.03E-09 (± 1.31E-07 ) | 3.72E-07 (± 5.45E-07 ) |
| terpinolene | 3.7E-07 (± 8.6E-07 ) | n.d. | 9.07E-09 (± 1.48E-07 ) | 5.77E-07 (± 8.77E-07 ) |
| linalool | 4.7E-07 (± 1.5E-06 ) | 2.8E-06 (± 4.7E-06 ) | 1.75E-07 (± 7.11E-07 ) | 2.46E-09 (± 1.85E-08 ) |
| myrcene | 1.5E-16 (± 4.7E-16 ) | n.d. | 4.69E-16 (± 2.72E-15 ) | 2.89E-16 (± 4.16E-16 ) |
| $\alpha$-terpineol[b] | 4.6E-09 (± 1.4E-08 ) | n.d. | n.d. | 3.87E-06 (± 5.98E-06 ) |
| longicyclene | 4.7E-08 (± 2.0E-07 ) | n.d. | n.d. | n.d. |
| isolongifolene and agurjunene | 9.78E-09 (± 2.99E-08 ) | 3.32E-09 (± 2.26E-08 ) | n.d. | n.d. |
| $\alpha$-humulene | 2.1E-06 (± 8.8E-06 ) | 4.03E-07 (± 6.82E-07 ) | 5.22E-07 (± 2.38E-06 ) | 2.31E-07 (± 3.78E-07 ) |
| $\beta$-farnesene | 1.1E-07 (± 3.2E-07 ) | 6.20E-08 (± 2.82E-07 ) | 9.88E-07 (± 3.82E-06 ) | 3.20E-07 (± 3.94E-07 ) |
| $\alpha$-farnesene[c] | 4.6E-07 (± 1.3E-06 ) | 1.66E-05 (± 2.53E-05 ) | 2.39E-05 (± 4.62E-05 ) | 4.80E-06 (± 2.61E-06 ) |
| $\beta$-caryophyllene | 3.0E-06 (± 1.2E-05 ) | 2.47E-06 (± 2.97E-06 ) | 1.45E-06 (± 5.27E-06 ) | 2.69E-09 (± 4.05E-08 ) |
| SQT1[d] | 9.8E-07 (± 4.0E-06 ) | 2.93E-07 (± 1.01E-06 ) | n.d. | 2.69E-08 (± 8.38E-08 ) |
| SQT2[d] | 1.2E-08 (± 4.3E-08 ) | n.d. | n.d. | n.d. |
| SQT3[d] | 2.7E-08 (± 1.1E-07 ) | n.d. | n.d. | n.d. |
| SQT5[e] | 3.9E-08 (± 1.8E-07 ) | n.d. | n.d. | n.d. |
| SQT6[d] | 3.7E-09 (± 1.7E-08 ) | n.d. | n.d. | 4.54E-07 (± 2.11E-06 ) |
| SQT7[d] | 5.3E-08 (± 1.3E-07 ) | n.d. | 5.15E-06 (± 1.06E-05 ) | n.d. |
| 1-hexanol | 3.7E-09 (± 6.7E-08 ) | 1.03E-09 (± 1.71E-08 ) | 3.68E-08 (± 1.85E-07 ) | 1.81E-08 (± 9.26E-08 ) |
| $cis$-2-hexen-1-ol | n.d. | n.d. | n.d. | n.d. |
| $trans$-2-hexen-1-ol | n.d. | 1.35E-07 (± 1.10E-06 ) | 1.13E-06 (± 4.51E-06 ) | 1.89E-07 (± 8.45E-07 ) |
| $cis$-3-hexen-1-ol | 6.6E-08 (± 7.5E-07 ) | 1.24E-06 (± 9.69E-06 ) | 8.59E-06 (± 3.13E-05 ) | 3.78E-06 (± 7.02E-06 ) |
| $trans$-3-hexen-1-ol | n.d. | n.d. | n.d. | n.d. |
| hexyl acetate | n.d. | n.d. | 7.50E-10 (± 1.22E-08 ) | n.d. |
| $cis$-3-hexenyl acetate | 1.1E-06 (± 8.1E-06 ) | 4.74E-06 (± 2.66E-05 ) | 2.56E-05 (± 8.64E-05 ) | 2.94E-06 (± 6.10E-06 ) |
| $trans$-2-hexenyl acetate | n.d. | 3.02E-08 (± 5.04E-07 ) | 5.57E-08 (± 6.42E-07 ) | n.d. |
| pentanal | 3.9E-08 (± 9.1E-08 ) | 3.78E-08 (± 7.49E-08 ) | 3.79E-08 (± 1.29E-07 ) | 1.97E-07 (± 8.46E-08 ) |





|  | Period B1 24 May–8 June | Period B2.a 21–29 June | Period B2.b 13–19 July | Period B3 23–28 August |
|---|---|---|---|---|
| hexanal | 5.8E-08 (± 1.6E-07 ) | 1.33E-07 (± 1.68E-07 ) | 2.68E-07 (± 6.37E-07 ) | 3.64E-07 (± 2.31E-07 ) |
| heptanal | 5.2E-08 (± 1.1E-07 ) | 2.70E-08 (± 1.05E-07 ) | 3.97E-08 (± 2.11E-07 ) | 1.87E-07 (± 1.46E-07 ) |
| octanal | 4.5E-08 (± 1.1E-07 ) | 4.71E-08 (± 1.51E-07 ) | 5.82E-08 (± 2.68E-07 ) | 2.49E-07 (± 2.00E-07 ) |
| nonanal | 5.6E-08 (± 1.3E-07 ) | 3.31E-07 (± 3.89E-07 ) | 2.05E-07 (± 5.79E-07 ) | 6.62E-07 (± 4.46E-07 ) |
| decanal | 1.4E-07 (± 3.3E-07 ) | 1.72E-07 (± 2.99E-07 ) | 8.81E-09 (± 7.21E-08 ) | 1.78E-07 (± 1.59E-07 ) |
| methacrolein | 3.1E-08 (± 7.2E-08 ) | 6.13E-08 (± 2.39E-07 ) | 4.64E-08 (± 6.86E-08 ) | n.d. |
| 1-pentanol | n.d. | n.d. | 3.57E-08 (± 3.74E-07 ) | n.d. |
| 1-octen-3-ol | n.d. | n.d. | n.d. | n.d. |
| butyl acetate | n.d. | n.d. | n.d. | n.d. |
| bornyl acetate | 1.1E-09 (± 3.6E-09 ) | 3.05E-09 (± 1.05E-08 ) | 3.79E-10 (± 6.18E-09 ) | 3.40E-10 (± 3.15E-09 ) |
| propanoic acid | 3.8E-07 (± 5.0E-07 ) | 2.25E-08 (± 4.88E-08 ) | 3.24E-08 (± 1.17E-07 ) | 3.88E-08 (± 9.76E-08 ) |
| butanoic acid | 4.7E-07 (± 3.9E-07 ) | 5.01E-08 (± 3.90E-08 ) | 1.93E-07 (± 1.90E-07 ) | 9.11E-08 (± 1.24E-07 ) |
| isobutanoic acid | 5.1E-09 (± 6.5E-08 ) | 8.36E-09 (± 4.43E-08 ) | 6.30E-09 (± 7.25E-08 ) | 4.39E-08 (± 1.22E-07 ) |
| pentanoic acid | 6.2E-09 (± 9.0E-08 ) | n.d. | n.d. | 1.50E-09 (± 2.26E-08 ) |
| isopentanoic acid | n.d. | n.d. | n.d. | n.d. |
| hexanoic acid | n.d. | n.d. | n.d. | n.d. |
| 4-methylpentanoic acid | n.d. | n.d. | n.d. | n.d. |

[a] quantified as carene    [b] quantified as terpinolene    [c] quantified as $\beta$-farnesene    [d] quantified as $\beta$-caryophyllene    [e] quantified as longicyclene





## Appendix E: COHRE and missing OHRE temperature dependence

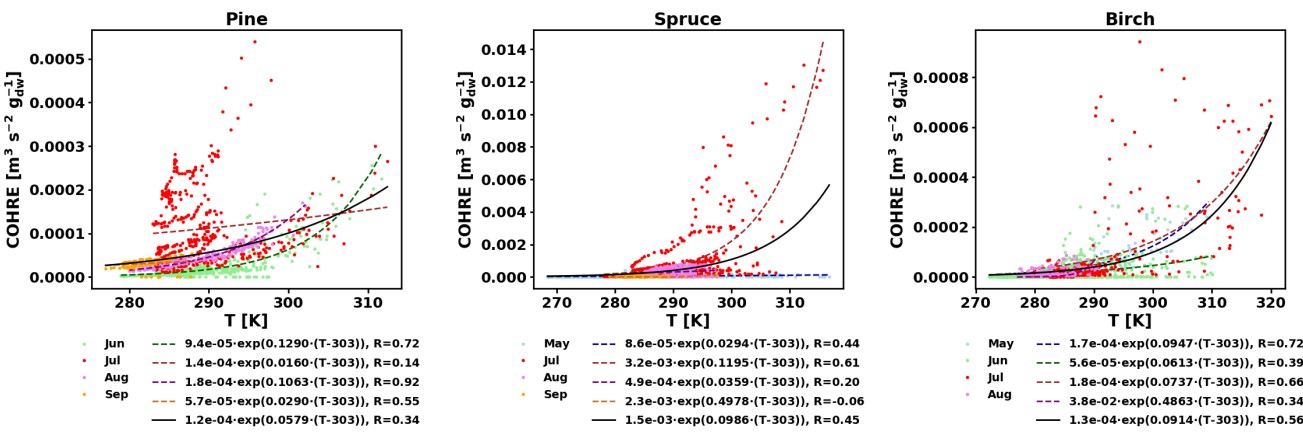

**Figure E1.** COHRE temperature dependence by month (coloured dots and dotted line fits) and fit for all data combined (black solid line) for pine (left), spruce (center), and birch (right).

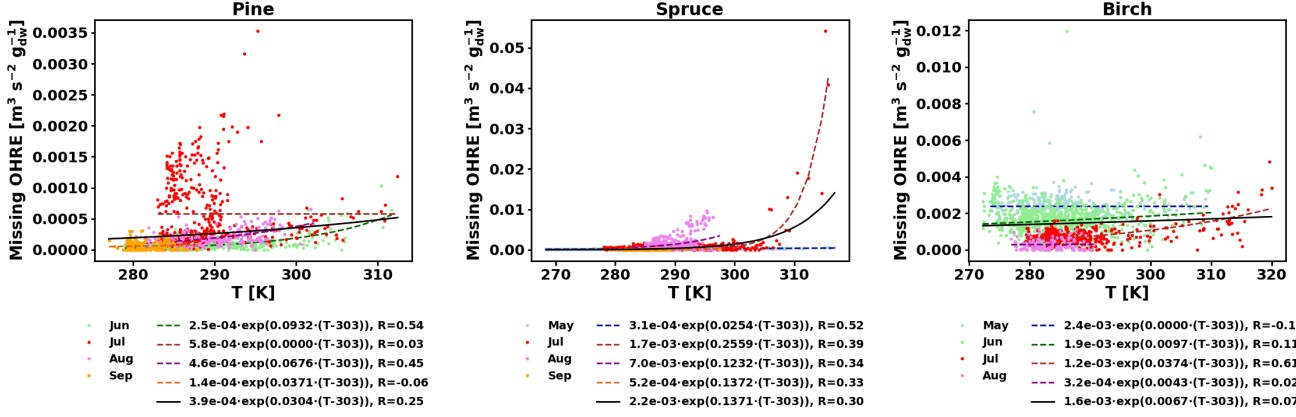

**Figure E2.** Missing OHRE temperature dependence by month (coloured dots and dotted line fits) and fit for all data combined (black solid line) for pine (left), spruce (center), and birch (right).





**Table E1.** Regression coefficients and coefficients of correlation ($R$) for temperature dependence of COHRE and for its dependence on both temperature and light using the hybrid algorithm.

| | Temperature dependence | | | Hybrid algorithm | | | |
|---|---|---|---|---|---|---|---|
| | $\text{COHRE}_s$ | $\beta$ | $R$ | $\text{COHRE}_{0,\text{pool}}$ | $\beta$ | $\text{COHRE}_{0,\text{synth}}$ | $R$ |
| | [$\text{m}^3\ \text{s}^{-2}\ \text{g}_{\text{dw}}^{-1}$] | [$\text{K}^{-1}$] | | [$\text{m}^3\ \text{s}^{-2}\ \text{g}_{\text{dw}}^{-1}$] | [$\text{K}^{-1}$] | [$\text{m}^3\ \text{s}^{-2}\ \text{g}_{\text{dw}}^{-1}$] | |
| **Pine** | | | | | | | |
| June | 9.4e-05 | 0.1290 | 0.72 | 9.4e-05 | 0.1290 | 1.4e-05 | 0.85 |
| July | 1.4e-04 | 0.0160 | 0.14 | 1.4e-04 | 0.0148 | 3.1e-03 | 0.12 |
| August | 1.8e-04 | 0.1063 | 0.92 | 1.8e-04 | 0.1065 | 9.8e-05 | 0.95 |
| September | 5.7e-05 | 0.0290 | 0.55 | 8.7e-02 | 0.4561 | 6.9e-01 | 0.27 |
| All | 1.2e-04 | 0.0579 | 0.34 | 1.2e-04 | 0.0579 | 1.8e-09 | 0.21 |
| **Spruce** | | | | | | | |
| May | 8.6e-05 | 0.0294 | 0.44 | 8.6e-05 | 0.0294 | 2.2e-07 | 0.24 |
| July | 3.2e-03 | 0.1195 | 0.61 | 3.2e-03 | 0.1195 | 7.5e-18 | 0.74 |
| August | 4.9e-04 | 0.0359 | 0.20 | 4.9e-04 | 0.0359 | 5.5e-08 | 0.19 |
| September | 2.3e-03 | 0.4978 | -0.06 | 8.7e-04 | 0.4996 | 1.1e+00 | 0.18 |
| All | 1.5e-03 | 0.0986 | 0.45 | 1.12e-03 | 0.0986 | 1.0e-18 | 0.24 |
| **Birch** | | | | | | | |
| May | 1.7e-04 | 0.0986 | 0.72 | 1.7e-04 | 0.0947 | 1.9e-05 | 0.80 |
| June | 5.6e-05 | 0.0613 | 0.39 | 5.6e-05 | 0.0613 | 4.8e-16 | 0.36 |
| July | 1.8e-04 | 0.0737 | 0.66 | 1.8e-04 | 0.0737 | 1.4e-06 | 0.65 |
| August | 3.8e-02 | 0.4863 | 0.34 | 3.6e-02 | 0.4872 | 3.1e-02 | 0.29 |
| All | 1.3e-04 | 0.0914 | 0.56 | 1.3e-04 | 0.0914 | 2.5e-08 | 0.36 |





**Table E2.** Regression coefficients and coefficients of correlation ($R$) for temperature dependence of missing OHRE (MOHRE) and for its dependence on both temperature and light using the hybrid algorithm.

| | Temperature dependence | | | Hybrid algorithm | | | |
|---|---|---|---|---|---|---|---|
| | MOHRE$_s$ | $\beta$ | $R$ | MOHRE$_{0,\mathrm{pool}}$ | $\beta$ | MOHRE$_{0,\mathrm{synth}}$ | $R$ |
| | [m$^3$ s$^{-2}$ g$_{\mathrm{dw}}^{-1}$] | [K$^{-1}$] | | [m$^3$ s$^{-2}$ g$_{\mathrm{dw}}^{-1}$] | [K$^{-1}$] | [m$^3$ s$^{-2}$ g$_{\mathrm{dw}}^{-1}$] | |
| **Pine** | | | | | | | |
| June | 2.5e-04 | 0.0932 | 0.54 | 2.5e-04 | 0.0932 | 7.3e-09 | 0.66 |
| July | 5.8e-04 | 0.0000 | 0.03 | 5.8e-04 | 0.0000 | 1.1e-23 | -0.03 |
| August | 4.6e-04 | 0.0676 | 0.45 | 4.6e-04 | 0.0676 | 7.9e-11 | 0.51 |
| September | 1.4e-04 | 0.0371 | -0.06 | 6.2e-05 | 0.0000 | 4.6e-11 | -0.12 |
| All | 3.9e-04 | 0.0304 | 0.25 | 3.9e-04 | 0.0304 | 2.4e-14 | 0.14 |
| **Spruce** | | | | | | | |
| May | 3.1e-04 | 0.0254 | 0.52 | 2.9e-04 | 0.0220 | 8.9e-03 | 0.49 |
| July | 1.7e-03 | 0.2559 | 0.39 | 1.7e-03 | 0.2559 | 6.0e-22 | 0.91 |
| August | 7.0e-03 | 0.1232 | 0.34 | 7.0e-03 | 0.1232 | 8.1e-13 | 0.37 |
| September | 5.2e-04 | 0.1372 | 0.33 | 2.9e-03 | 0.4981 | 1.1e+00 | 0.27 |
| All | 2.2e-03 | 0.1371 | 0.30 | 2.2e-03 | 0.1371 | 2.2e-37 | 0.40 |
| **Birch** | | | | | | | |
| May | 2.4e-03 | 0.0000 | -0.11 | 2.4e-03 | 0.0000 | 6.7e-04 | 0.00 |
| June | 1.9e-03 | 0.0097 | 0.11 | 1.5e-03 | 0.0000 | 2.2e-01 | 0.26 |
| July | 1.2e-03 | 0.0374 | 0.61 | 1.2e-03 | 0.0373 | 2.6e-04 | 0.60 |
| August | 3.2e-04 | 0.0043 | 0.02 | 2.8e-04 | 0.0000 | 3.7e-01 | 0.06 |
| All | 1.6e-03 | 0.0067 | 0.07 | 1.4e-03 | 0.0000 | 1.7e-01 | 0.20 |





*Author contributions.*

A. P. Praplan conducted total OH reactivity measurements, performed data analysis, and lead the writing of the manuscript. T. Tykkä operated GC-MSs and analysed the data produced. S. Schallhart participated in the data analysis and commented on the manuscript. J. Bäck assisted in the interpretation of the results and gave comments on the manuscript. H. Hellén designed the study, conducted GC-MSs measurements, analysed the data and commented on the manuscript.

*Acknowledgements.* The presented research has been funded by the Academy of Finland (Academy Research Fellowship, projects nos. 307797, 275608 and 312502, as well as Centre of Excellence in Atmospheric Science, grant no. 272041). The authors thank Hannele Hakola for the continuous support. They also thank the staff at the SMEAR II station for their help, Dr. Jari Waldén for lending calibration standards.



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
