# Peer review of "OH reactivity from different tree species: Investigating the missing reactivity in a boreal forest"

_Biogeosciences, 2020_

## Referee Comment (RC1) · Anonymous Referee #1 · 17 Mar 2020

The manuscript "OH reactivity from different tree species: Investigating the missing reactivity in a boreal forest" by Praplan et al. presents results of emission measurements of VOC and the total OH reactivity from three typical boreal tree species over a time period from May to September. The variations in the total OH reactivity emissions (TOHRE) between tree species, seasons, temperature and light are discussed. A comparison to individually detected VOC emissions, by multiplying the measured concentrations with their reaction rates with OH and summing up, revealed a highly variable missing gap. This gap shows that the direct tree emissions were not fully determined by the typically applied methods such as gas chromatography.

[Figure]

The general topic further fuels the discussion of where the missing OH reactivity in forests originates from and is of high interest for understanding the processes of both biogenic emissions and atmospheric chemistry. The dataset seems of high quality, is presented thoroughly and discussed in many aspects. I find that especially an extended discussion about uncertainties and detection limits could improve the current state of the manuscript significantly. Therefore, I recommend this manuscript for publication after addressing the following specific comments:
* * *
Specific comments:

This manuscript has a focus on measurements of total OH reactivity emissions from three different boreal tree species. I wonder why the key-word "emissions" is not part of the title.

p.2, l.29-35: In this paragraph the concept of total OH reactivity is introduced from a historical point of view. It is described that the motivation to determine the total OH reactivity is the inaccessibility of "OH sinks in the model". For the reader to understand the following paragraphs, an accurate definition of the total OH reactivity would be needed here.

p.2, l. 47: This paragraph introduces previous results of measurements of the total OH reactivity to the reader. It is said that the studies of Sinha et al. (2010) and Nölscher et al. (2012) find missing reactivity in the boreal forest. Then you refer to Praplan et al. (2019) who "recently demonstrated that including modelled oxidation products of VOCs that are not measured is not sufficient to explain the missing reactivity at the site". Please, make clear what site you are talking about here.

p.2, l.53/54: The study of Nölscher et al. (2013) examined TOHRE of Norway Spruce, not Scots Pine.

p.3, l.73: In the section 2.2 the studied tree species are described. Seedlings of three

boreal tree species were chosen for enclosing branches and detecting the branch emissions. How representative are the emissions of seedling? How comparable are the emissions of a "young" tree with an "old-grown" one?

p.4, l.91: The study was conducted with three branch enclosures, which were placed on three different trees. As I understand, after some time the enclosures had to be moved and another branch of the tree was enclosed. Over the studied period, each tree was subsequently sampled on three branches. Do you have evidence, that this method provides comparable results? I wonder, did you (or another study) test to measure the emission of various tree branches simultaneously and found that they provide the same results?

p.4, l.98: How were the data treated when the temperature differences were high? Was a threshold value defined to filter out high-temperature, hence unrealistic, data?

p.4, l.100: In the section 2.4 the VOC-measurements via gas chromatography are described. These measurements are vital for determining the OH reactivity fraction that can be explained or, when subtracted from the measured TOHRE, the missing OHRE. Therefore, please describe here the calibration method used, the uncertainty of the measurement and the limit of detection.

p.5, l.138: The reaction rate of pyrrole with OH was determined recently by Dillon et al. (2012).

p.5, l.122: The Comparative Reactivity Method (CRM) is used here for detecting the total OH reactivity with a gas chromatograph equipped with a PID. What is the uncertainty of this instrument and the limit of detection? Do you find interfering compounds in the short time period of the two-minute chromatogram?

p.5, l.144: Here it is described how the total OH reactivity background was experimentally derived by measuring from an empty enclosure. In light of the following results and discussion, I think it is necessary to determine a detection limit for the TOHRE

measurements. With the noise of the blank enclosure measurement, a 2-sigma value for the total OH reactivity can be calculated and with this a detection limit for TOHRE. Especially, when comparing the measured and calculated OHRE during low emission periods, this value will aid to judge the significance of the results.

p.8, l.205: How do the measured TOHRE values compare to previous studies (e.g. Nölscher et al. 2013)?

p.9, l.218/219: The "deviation from pseudo first-order kinetics applied to the CRM data is based on calibration with a-pinene as a surrogate for biogenic emissions, but monoterpenes do not always represent the largest fraction of the emissions, which result in some uncertainty in TOHRE." This uncertainty could be quantified by comparing the ka-pin with a reaction rate that represents the measured VOCs.

p.9, l.222-226: The authors discuss here the results that will be shown in the following paragraphs. This is difficult to follow by the reader.

p.9, l.227-p.15, l.295: Sections 3.2 to 3.4 present and discuss the results, however do not refer at any time to the pictured figures. The reader has to guess what to look at. Additionally, the periods indicated in the figures and discussed in the text are not introduced to the reader. How are these periods defined? Why do you not show simultaneous data periods between the three trees?

p.16, l.300: "Good correlations with temperature are found for the TOHRE ...". Is it really a linear dependency or an exponential dependency? Here the text is rather vague what type of regression was used. Only in the caption of Figure 5 it is stated that an exponential regression is calculated for the temperature dependency of THORE. Please clarify (here and also in the caption of Figure 5 and Table 2).

p.16, l.322: The last sentence mentions other factors that can play a major role on the type and amount of reactive emissions. Please, provide examples of that factors. Can you see the effect of these factors in your dataset?

Technical comments:

While the manuscript overall is well structured and clearly written, it should be checked for the English language. There are a number of language mistakes, that I spotted and probably more than that could be improved in the language.

Examples: p.1, l.2: missing s at the end of a verb . . . a large fraction of total hydroxyl (OH) reactivity remains. . .. p. 2, l.42: unnecessary closing bracket . . . during which the forest experienced stressed conditions) . . . p.5, l.126: missing the . . .and used for the rest of the measurement periods. p.8, l.212: missing s for plural . . . in large amounts. . . p.9, l.226: conjugation of the verb to past tense . . .cannot be explained only by. . . p.12, l.252: verb used in singular form when needed in plural . . . values are observed. . .

Figures 2, 3 and 4:

The main results are presented in the Figures 2, 3 and 4. The upper parts of the figures show the measured TOHRE as well as COHRE (the calculated OH reactivity emissions). This should be indicated in the legend.

The scatter of TOHRE is especially for periods of low emissions high. Please indicate here the lower detection limit of TOHRE. Which emissions of total OH reactivity can be reliably identified with the method? When are emissions too low? In that case, the missing OHRE should be treated with care.

Can you provide and include the uncertainty of COHRE into the figure?

Maybe the figure would be easier to read if pictured across the entire page when turned about 90°.

p.16, l. 305-310: For consistency with the methods part of this manuscript use Kelvin in the units of the beta-factor.

Reference:

Dillon et al. (2012): Terry J. Dillon, Maria E. Tucceri, Katrin Dulitz, Abraham Horowitz, Luc Vereecken, and John N. Crowley: Reaction of Hydroxyl Radicals with C4H5N (Pyrrole): Temperature and Pressure Dependent Rate Coefficients, The Journal of Physical Chemistry A 2012 116 (24), 6051-6058, DOI: 10.1021/jp211241x.

---

## Referee Comment (RC2) · Anonymous Referee #2 · 12 Apr 2020

This manuscript is an important contribution to our knowledge about the emission of BVOCs from vegetation, particularly from boreal forest, and its implication for OH reactivity, considering current gaps in our knowledge in this research area. In particular, this work highlights some important aspects related to difficulties in OH reactivity evaluation, and its dependence on both environmental conditions and methodological limitations. My comments are mostly about the way the results are presented and discussed. While the methodological aspects are generally described in detail, in few cases important aspects should be clarified. There are a number of language flaws that should be corrected. Specific comments are listed below.

[Figure]

Specific comments:

Line 11: why "even though"?

Introduction: While you emphasize the importance of accounting for OH reactivity by its effects on the lifetime of VOCs oxidants in the atmosphere, there are additional central reasons why OH reactivity and BVOCs composition are important. Among these you can mention photochemical air-pollution and secondary aerosol formation.

Line 29-35: This paragraph use extensively "OH reactivity" which you define on line 111. It will be good to provide a brief explanation/definition before you discuss it here.

Line 31: Can you provide reference/s to support this statement?

Line 42: remove extra ")".

Line 50: "contradictory" – the use of this word seems inappropriate taking into account the rest of the paragraph.

Line 57: What do you mean by "important" trees? Can you make this point clearer?

Line 83: It's not a comparison between the "changes in biomass" and other uncertainties; you probably refer to the effect emanating from the former on OH reactivity. While this assumption seems reasonable can you indicate to what extent (e.g., in percentage) this effect could affect your results?

Lines 141-142: Can you explain why you didn't use C3H8 for the calibration as described by Parplan et al. (2019) and Shina et al. (2008)?

Line 147: Are the values in Eq. 3 referring to the calibration factors? Please indicate their meaning in the text.

Line 148: To what extent humidity can affect your results in addition to (/relative to) the dilution effect?

Line 152: "Other correction factors need to be applied..." – Do you mean other than

the presence of NOX and O3? If so, why did you write "However" in the next sentence?

Lines 152-153: Is this because NOX and O3 are assumed to be effectively removed by the HPZA-7000?

Line 164: "OH levels" – do you mean the effect on RH levels?

Line 168: "t o" should be "to".

Line 180: Eq. 8, please indicate clearly what is the meaning of the (calibration?) values.

Lines 187-188: Please provide a reference to support this estimation.

Results and discussion: In this section you present 4 figures and 2 tables, but you only once refer to Fig. 5 and Table 1 and 2. Please try to refer the reader to each of these from the text.

Lines 209-2013: I suggest making this paragraph shorter and generally avoid summarizing or discussing the results prior presenting the results themselves.

Table 1 caption: Acronyms - "Te" is not consistent with RH if the latter refers to measurements in the enclosure.

Line 214: Can you support it by statistical analysis?

Line 235: GLVs was already defined before.

Line 236: "as well" appears twice.

Lines 245-247: Can you elaborate on why the fact that stress in your case was not driven by elevated temperature indicates that some of these (missing OHRE) are not terpenoids or oxidized volatile organic compounds?

Line 249: What do you mean by "follows qualitatively"? If you imply for correlation, can you calculate the Pearson correlation coefficient and present it to support this notion?
Lines 253-255: What about aldehydes? How it compares with the findings by Hakola et al. (2017) about aldehyde emissions from Spruce?

Lines 255-257: Does the higher emission of sesquiterpenes at the expense of GLVs under the different conditions can be supported by statistical analysis? Can you use regression to identify which of the parameters (temperature, radiation, precipitation) shows higher association with the emissions rate of the various VOCs?

Line 259: Why "However?

Lines 280-281: Can you indicate to what extent the constant blank subtraction could have affected your results?

Line 282: "quantitatively" - Qualitatively? Can you calculate the Pearson correlation coefficient for all three species?

Line 300: Please indicate the regression you have used to evaluate R. Did you try exponential regression (as is implied by Fig. 5)?

Lines 301-303: Can you provide more information about the nature of the stress and the cause for the low TOHRE?

Line 305: Why not referring to Table 2?

Line 312: What do you mean by "good correlation with temperature"?

Line 316: "In a few cases was even slightly reduced." – this seems reasonable to me. I just want to make sure you have used exponential regression type for temperature.

Lines 321-323: A very general sentence - can you specify which factors ("other factors") and elaborate on that?

Line 332: "stress-induced" - Please specify stress type as much as possible.

Lines 337-338: Can you provide an explanation for that?

Line 341: "highest" - Looks like a contradiction with the rest of the sentence.

References

Hakola, H., Tarvainen, V., Praplan, A. P., Jaars, K., Hemmilä, M., Kulmala, M., Bäck, J., and Hellén, H.: Terpenoid and carbonyl emissions from Norway spruce in Finland during the growing season, Atmos. Chem. Phys., 17, 3357–3370, https://doi.org/10.5194/acp-17-3357-2017, 2017.

Praplan, A. P., Tykkä, T., Chen, D., Boy, M., Taipale, D., Vakkari, V., Zhou, P., Petäjä, T., and Hellén, H.: Long-term total OH reactivity measurements in a boreal forest, Atmos. Chem. Phys., 19, 14431–14453, https://doi.org/10.5194/acp-19-14431-2019, 2019.

Sinha, V., Williams, J., Crowley, J. N., and Lelieveld, J.: The Comparative Reactivity Method – a new tool to measure total OH Reactivity in ambient air, Atmos. Chem. Phys., 8, 2213–2227, https://doi.org/10.5194/acp-8-2213-2008, 2008.

---

## Author Comment (AC1) · 29 May 2020

We would like to first thank the referee for acknowledging the importance of our study. We are grateful for their comments, which help us clarify the way we presented and discussed our results. We believe that the revised manuscript can address the referee's concerns and has improved due to the referee's input. In the following, the referee's comments are marked in italics and our answers are written with the regular font.

*The manuscript "OH reactivity from different tree species: Investigating the missing reactivity in a boreal forest" by Praplan et al. presents results of emission measurements of VOC and the total OH reactivity from three typical boreal tree species over*

*a time period from May to September. The variations in the total OH reactivity emissions (TOHRE) between tree species, seasons, temperature and light are discussed. A comparison to individually detected VOC emissions, by multiplying the measured concentrations with their reaction rates with OH and summing up, revealed a highly variable missing gap. This gap shows that the direct tree emissions were not fully determined by the typically applied methods such as gas chromatography.*

*The general topic further fuels the discussion of where the missing OH reactivity in forests originates from and is of high interest for understanding the processes of both biogenic emissions and atmospheric chemistry. The dataset seems of high quality, is presented thoroughly and discussed in many aspects. I find that especially an extended discussion about uncertainties and detection limits could improve the current state of the manuscript significantly. Therefore, I recommend this manuscript for publication after addressing the following specific comments:*

*Specific comments:*

*This manuscript has a focus on measurements of total OH reactivity emissions from three different boreal tree species. I wonder why the key-word "emissions" is not part of the title.*

It is indeed only implicit in the original title of the manuscript that we are investigating emissions. The revised manuscript's title now states it explicitly: "OH reactivity from the emissions of three different tree species: Investigating the missing reactivity in a boreal forest".

*p.2, l.29-35: In this paragraph the concept of total OH reactivity is introduced from a historical point of view. It is described that the motivation to determine the total OH reactivity is the inaccessibility of "OH sinks in the model". For the reader to understand the following paragraphs, an accurate definition of the total OH reactivity would be needed here.*

We edited this paragraph to define total OH reactivity as the "total OH loss rate". We also state that it can be seen as the inverse of OH lifetime, meaning that high OH reactivity values translate into short atmospheric OH lifetimes (see also comment from Anonymous Referee 2).

*p.2, l. 47: This paragraph introduces previous results of measurements of the total OH reactivity to the reader. It is said that the studies of Sinha et al. (2010) and Nölscher et al. (2012) find missing reactivity in the boreal forest. Then you refer to Praplan et al. (2019) who "recently demonstrated that including modelled oxidation products of VOCs that are not measured is not sufficient to explain the missing reactivity at the site". Please, make clear what site you are talking about here.*

We have clarified in the revised manuscript that all these studies have been performed at the SMEAR II boreal forest station in Hyytiälä, Finland.

*p.2, l.53/54: The study of Nölscher et al. (2013) examined TOHRE of Norway Spruce, not Scots Pine.*

We regret to have let this mistake in the introduction. It has been corrected in the revised manuscript. We, however, discussed correctly the results of Nölscher et al. (2013) for Norway spruce in the discussion section and cited it in the appropriate context in the abstract.

*p.3, l.73: In the section 2.2 the studied tree species are described. Seedlings of three boreal tree species were chosen for enclosing branches and detecting the branch emissions. How representative are the emissions of seedling? How comparable are the emissions of a "young" tree with an "old-grown" one?*

We perfectly understand the concerns of the referee. However, we do not claim that our study yields results that can be upscaled immediately. It is a part of a more global conversation regarding BVOCs emissions in the boreal forest.

While emissions vary with the seasons, the environmental conditions and even from

tree to tree of the same species (chemotypes; Bäck et al., 2012), often studies made of various trees (e.g. age, location) of the same species yield similar results. Also in our study we cite earlier work where similar emission patterns were measured in the discussion section.

The choice of using seedlings was mostly practical as it allowed to bring the trees close to the container with the instrumentation, which can be an advantage over building long sampling line (or move the container, which might prove difficult).

We expanded the rationale for the use of seedlings in section 2.2, while acknowledging that they cannot be considered representative without a borader context of emission measurements. We added the following: "The use of seedlings in pots was mostly practical as it was easier to bring them close to the instruments that characterise the emissions; moving the instruments' container closer to the trees of interest is not possible. Additionally, extremely long sampling lines and wall losses could be avoided. Emissions from the seedlings might not be representative per se. Nevertheless, put in perspective with results from other studies, they provide valuable information for any potential upscaling effort."

*p.4, l.91: The study was conducted with three branch enclosures, which were placed onthree different trees. As I understand, after some time the enclosures had to be moved and another branch of the tree was enclosed. Over the studied period, each tree was subsequently sampled on three branches. Do you have evidence, that this method provides comparable results? I wonder, did you (or another study) test to measure the emission of various tree branches simultaneously and found that they provide the same results?*

While we had three branch enclosures available, we had two GC instrument measuring each different compounds and only one instrument to measure total OH reactivity. Simultaneous measurements were therefore not possible in this study. Bertin et al. (1997) showed that branch-to-branch variability (for sun-exposed branches exposed to

sunlight) is similar to tree-to-tree variability for the evergreen oak. However, a large difference (190 %) was observed between sun-exposed branches and shade-adapted branch. In our study, the branches are both exposed to sunlight and in the shadow, depending on the time of the day. Also, as stated previously, emissions from this study (and others) show similar emissions patterns for a given tree species. Taken all together these studies contribute to identify the potential causes of variations.

*p.4, l.98: How were the data treated when the temperature differences were high? Was a threshold value defined to filter out high-temperature, hence unrealistic, data?*

It is a known issue of branch enclosures that the temperature in the enclosure exceeds ambient temperature. By removing data above 30 °C, changes for $\beta$ factors when the coefficient of correlation R is larger than 0.5 are within 15 %, except for spruce data in July (highest temperatures), where the $\beta$ factor (0.1853 without filtering data) decrease to 0.0931. We included this information in the revised manuscript.

*p.4, l.100: In the section 2.4 the VOC-measurements via gas chromatography are described. These measurements are vital for determining the OH reactivity fraction that can be explained or, when subtracted from the measured TOHRE, the missing OHRE. Therefore, please describe here the calibration method used, the uncertainty of the measurement and the limit of detection.*

The instrument was calibrated for MBO, aldehydes, mono- and sesquiterpenes using liquid standards in methanol solutions. Isoprene was calibrated using a gaseous standard (National Physical Laboratory, 32 VOC mix at 4ppb level). Limits of detections for mono- and sesquiterpenes are comprised between 0.5 and 4.7 pptv and the uncertainty of the measurements lies at 17–20 % (Helin et al., 2020). For organic acids, the detection limits are in 1–130 pptv and the uncertainty is 32–76 % (Hellén et al., 2017). For other compounds that had no standard available, the uncertainty and the detection limits were estimated based on similar compounds. We included this information in the revised manuscript.

*p.5, l.138: The reaction rate of pyrrole with OH was determined recently by Dillon et al. (2012).*

We thank the referee for pointing our attention to this more recent measurement of the reaction rate of pyrrole with OH. The reaction rate used in this study ($1.2 \cdot 10^{-10}$ cm$^3$s$^{-1}$, Atkinson et al., 1985) is, nevertheless, within the uncertainty of the value determined by Dillon et al., ($1.28 \pm 0.1 \cdot 10^{-10}$ cm$^3$s$^{-1}$). While we did not discussed in detail uncertainty calculations in the original manuscript, we did in our previous publications (Praplan et al., 2017; 2019), where we state that an uncertainty of 15 % is used for this reaction rate.

*p.5, l.122: The Comparative Reactivity Method (CRM) is used here for detecting the total OH reactivity with a gas chromatograph equipped with a PID. What is the uncertainty of this instrument and the limit of detection? Do you find interfering compounds in the short time period of the two-minute chromatogram?*

We added here information about the GC-PID. Its uncertainty is about 5 % and its limit of detection is 1.7 ppbv ($2\sigma$). The retention time of pyrrole is roughly 65 s and we never observed interfering compounds. When the pyrrole flow to the instrument is switched off under various conditions, no peak is ever observed at the position of the pyrrole peak. This is now also explicitly mentioned in the revised manuscript.

*p.5, l.144: Here it is described how the total OH reactivity background was experimentally derived by measuring from an empty enclosure. In light of the following results and discussion, I think it is necessary to determine a detection limit for the TOHRE measurements. With the noise of the blank enclosure measurement, a 2-sigma value for the total OH reactivity can be calculated and with this a detection limit for TOHRE. Especially, when comparing the measured and calculated OHRE during low emission periods, this value will aid to judge the significance of the results.*

We thank the referee for this helpful suggestion. In addition of subtracting blank values, we plot the $2\sigma$ value for total OH reactivity from the blank cuvette adjusted for flow

through the enclosure and dry weight of the needle/leaves and plot it in the top row of Figures 2 to 4 to indicate the limit of detection of the method. In addition the missing fraction is displayed now only for cases when the measured total OH reactivity is higher than the limit of detection (defined as $2\sigma$ of the total OH reactivity from the blank measurement in an empty enclosure). This did not affect significantly the monthly averages in Table 1.

*p.8, l.205: How do the measured TOHRE values compare to previous studies (e.g. Nölscher et al. 2013)?*

We would have liked to compare our measurements to the previous studies. However, in Nölscher et al. (2013), the "Total OH reactivity emission rates were expressed as emitted total OH reactivity ($R_{\text{total}} in s^{-1}$) per unit needle dry weight (g(dw)$^{-1}$) per unit enclosure volume (m$^{-3}$) per unit time (s$^{-1}$)", resulting in s$^{-2}$ g(dw)$^{-1}$ m$^{-3}$ as TOHRE units, while our study uses a similar normalization as for VOC emissions (Eq. 5), resulting in s$^{-2}$ g(dw)$^{-1}$ m$^3$ units for TOHRE.

This difference seems to stem from the different method used: dynamic branch enclosure in Nölscher et al. (2013) and flow through technique in our study. We failed to see how to reconcile the units and therefore refrained from comparing values and opted for a qualitative comparison only. We added this information to the revised manuscript.

*p.9, l.218/219: The "deviation from pseudo first-order kinetics applied to the CRM data is based on calibration with a-pinene as a surrogate for biogenic emissions, but monoterpenes do not always represent the largest fraction of the emissions, which result in some uncertainty in TOHRE." This uncertainty could be quantified by comparing the ka-pin with a reaction rate that represents the measured VOCs.*

For reactivity calibrations with propane from our earlier work, the slope of the regression comparing measured reactivity ($R_{eqn}$) and expected reactivity ($R_{\text{true}}$) is 0.751. We do not have reactivity calibrations for compounds with higher reactivity than monoterpenes (e.g. sesquiterpenes), but based on reaction rate coefficients, the difference

in reactivity between propane and $\alpha$-pinene is larger to the difference in reactivity between $\alpha$-pinene and $\beta$-cayophyllene. Therefore, considering the uncertainty between the slopes of the regressions for propane and $\alpha$-pinene (51 %, lower uncertainty), it is reasonable to assume a lower upper uncertainty. We extended our statement regarding the uncertainty of this correction in the revised manuscript.

*p.9, l.222-226: The authors discuss here the results that will be shown in the following paragraphs. This is difficult to follow by the reader.*

We acknowledge that this was not the right place to discuss the results showed only later. This has been corrected in the revised manuscript by moving the paragraph to the end of this section and rewriting parts of this section (see also Anonymous Referee 2's comment).

*p.9, l.227-p.15, l.295: Sections 3.2 to 3.4 present and discuss the results, however do not refer at any time to the pictured figures. The reader has to guess what to look at. Additionally, the periods indicated in the figures and discussed in the text are not introduced to the reader. How are these periods defined? Why do you not show simultaneous data periods between the three trees?*

We regret that we were not able to present our data in the most intelligible fashion. In the revised manuscript we have partly rewritten sections 3.2 to 3.4 with the proper references to the various figures and with a proper definition of the periods indicated in the figures in the main text and not only in the caption of the figures. As we only have one CRM instrument available, we are unable to measure several trees simultaneously.

*p.16, l.300: "Good correlations with temperature are found for the TOHRE...". Is it really a linear dependency or an exponential dependency? Here the text is rather vague what type of regression was used. Only in the caption of Figure 5 it is stated that an exponential regression is calculated for the temperature dependency of THORE. Please clarify (here and also in the caption of Figure 5 and Table 2).*

We did reference Eq. (9) in the main text of the original manuscript. We now mention explicitly in the previous paragraph that we use it for exponential regressions in the revised manuscript. The caption of Figure 5 included already twice the expression "exponential regression" in the original manuscript, therefore we did not modify it, but we updated Table 2 to emphasize this important aspect of our study.

*p.16, l.322: The last sentence mentions other factors that can play a major role on the type and amount of reactive emissions. Please, provide examples of that factors. Can you see the effect of these factors in your dataset?*

We referred to abiotic stress factors as we see in particular in the data for pine and to some degree for spruce with high TOHRE values at moderate temperatures. We mention this explicitly as an example in the revised manuscript (see also answer to comment by Anonymous Referee 2).

*Technical comments:*

*While the manuscript overall is well structured and clearly written, it should be checked for the English language. There are a number of language mistakes, that I spotted and probably more than that could be improved in the language.*

*Examples: p.1, l.2: missing s at the end of a verb...a large fraction of total hydroxyl(OH) reactivity remains.... p. 2, l.42: unnecessary closing bracket...during which the forest experienced stressed conditions)...p.5, l.126: missing the...and used for the rest of the measurement periods. p.8, l.212: missing s for plural...in large amounts...p.9, l.226: conjugation of the verb to past tense...cannot be explained only by...p.12,l.252: verb used in singular form when needed in plural...values are observed...*

The mistakes pointed out by the referee (and others) have been corrected and the manuscript has been submitted to a professional language check (see also comments by Anonymous Referee 2).

*Figures 2, 3 and 4:*

*The main results are presented in the Figures 2, 3 and 4. The upper parts of the figures show the measured TOHRE as well as COHRE (the calculated OH reactivity emissions). This should be indicated in the legend.*

We have replaced the coloured surface depicting COHRE in the upper parts of the figures with a solid red line and indicate COHRE in the legend as well. In addition, the y-axis label has been renamed to "OHRE" instead of "TOHRE".

*The scatter of TOHRE is especially for periods of low emissions high. Please indicate here the lower detection limit of TOHRE. Which emissions of total OH reactivity can be reliably identified with the method? When are emissions too low? In that case, the missing OHRE should be treated with care.*

We indicate now in Figures 2 to 4 the limit of detection based on $2\sigma$ of the signal in an empty enclosure (blank, see also answer to earlier comment). Emissions are low usually during the night and the high missing fraction results indeed mostly from these low TOHRE values, close to the detection limit and these results should be treated with caution.

*Can you provide and include the uncertainty of COHRE into the figure? Maybe the figure would be easier to read if pictured across the entire page when turned about 90âŮę.*

COHRE is derived from up to 67 compounds. Considering the uncertainty from the GC-MS measurements and on the reaction rates used to derive COHRE, the uncertainty on COHRE could be up to a factor 4.3. Nevertheless, not all 67 compounds are measured simultaneously for all three trees and all measurement periods. We therefore estimated the uncertainty of COHRE for each measurement point individually according to the compounds contributing to it and display it now in the figure in the revised manuscript. It is mostly comprised between 25 and 50 %. This information has been included in the main text of the revised manuscript.

*p.16,l.305-310: For consistency with the methods part of this manuscript use Kelvin in the units of the beta-factor.*

Indeed, we should have used Kelvin as a unit for consistency. This has been fixed in the revised manuscript.

*References*

Atkinson, R., Aschmann, S. M., Winer, A. M., and Carter, W. P. L.: Rate constants for the gas-phase reactions of nitrate radicals with furan, thiophene, and pyrrole at 295 $\pm$ 1 K and atmospheric pressure, Environ. Sci. Technol., 19, 87–90, 1985. doi:10.1021/es00131a010.

Bäck, J., Aalto, J., Henriksson, M., Hakola, H., He, Q., and Boy, M.: Chemodiversity of a Scots pine stand and implications for terpene air concentrations, Biogeosciences, 9, 689–702, 2012. doi:10.5194/bg-9-689-2012.

Bertin, N., Staudt, M., Hansen, U., Seufert, G., Ciccioli, P., Foster, P., Fugit, J. L. and Torres, L.: Diurnal and seasonal course of monoterpene emissions from Quercus ilex (L.) under natural conditions application of light and temperature algorithms, Atmos. Env., 31, 135–144, 1997. doi:10.1016/S1352-2310(97)00080-0.

*Dillon et al. (2012): Terry J. Dillon, Maria E. Tucceri, Katrin Dulitz, Abraham Horowitz, Luc Vereecken, and John N. Crowley: Reaction of Hydroxyl Radicals with C4H5N (Pyrrole): Temperature and Pressure Dependent Rate Coefficients, The Journal of Physical Chemistry A, 116, 6051-6058, 2012. doi: 10.1021/jp211241x.*

Helin, A., Hakola, H., and Hellén, H.: Development of a thermal desorption-gas chromatography-mass spectrometry method for the analysis of monoterpenoids, sesquiterpenoids and diterpenoids, Atmos. Meas. Tech. Discuss., in review, 2020. doi:10.5194/amt-2019-469.

Hellén, H., Schallhart, S., Praplan, A. P., Petäjä, T., and Hakola, H.: Using in situ GC-MS for analysis of C2–C7 volatile organic acids in ambient air of a boreal forest site,

Atmos. Meas. Tech., 10, 281–289, 2017. doi:10.5194/amt-10-281-2017.

*Nölscher, A. C., Williams, J., Sinha, V., Custer, T., Song, W., Johnson, A. M., Axinte, R., Bozem, H., Fischer, H., Pouvesle, N., Phillips, G., Crowley, J. N., Rantala, P., Rinne, J., Kulmala, M., Gonzales, D., Valverde-Canossa, J., Vogel, A., Hoffmann, T., Ouwersloot, H. G., Vilà-Guerau de Arellano, J., and Lelieveld, J.: Summertime total OH reactivity measurements from boreal forest during HUMPPA-COPEC 2010, Atmos. Chem. Phys., 12, 8257–8270, 2012. doi:10.5194/acp-12-8257-2012.*

*Nölscher, A. C., Bourtsoukidis, E., Bonn, B., Kesselmeier, J., Lelieveld, J., and Williams, J.: Seasonal measurements of total OH reactivity emission rates from Norway spruce in 2011, Biogeosciences, 10, 4241–4257, 2013. doi:10.5194/bg-10-4241-2013.*

Praplan, A. P., Pfannerstill, E. Y., Williams, J., and Hellén, H.: OH reactivity of the urban air in Helsinki, Finland, during winter, Atmos. Environ., 169, 150 – 161, 2017. doi:10.1016/j.atmosenv.2017.09.013.

*Praplan, A. P., Tykkä, T., Chen, D., Boy, M., Taipale, D., Vakkari, V., Zhou, P., Petäjä, T.,and Hellén, H.: Long-term total OH reactivity measurements in a boreal forest, Atmos.Chem. Phys., 19, 14431–14453, 2019. doi:10.5194/acp-19-14431-2019.*

*Sinha, V., Williams, J., Lelieveld, J., Ruuskanen, T., Kajos, M., Patokoski, J., Hellen, H., Hakola, H., Mogensen, D., Boy, M., Rinne, J., and Kulmala, M.: OH Reactivity Measurements within a Boreal Forest: Evidence for Unknown Reactive Emissions, Environ. Sci. Technol., 44, 6614–6620, 2010. doi:10.1021/es101780b.*

---

## Author Comment (AC2) · 29 May 2020

We would like to first thank the referee for acknowledging the importance of our study. We are grateful for their comments, which help us clarify the way we presented and discussed our results. We believe that the revised manuscript can address the referee's concerns and has improved due to the referee's input. In the following, the referee's comments are marked in italics and our answers are written with the regular font.

*This manuscript is an important contribution to our knowledge about the emission of BVOCs from vegetation, particularly from boreal forest, and its implication for OH reactivity, considering current gaps in our knowledge in this research area. In particu-*

*lar, this work highlights some important aspects related to difficulties in OH reactivity evaluation, and its dependence on both environmental conditions and methodological limitations. My comments are mostly about the way the results are presented and discussed. While the methodological aspects are generally described in detail, in few cases important aspects should be clarified. There are a number of language flaws that should be corrected. Specific comments are listed below.*

We thank the referee for his comments and for pointing out the sentences in which we were not able to express clearly what we wanted to tell. The revised manuscript underwent a professional language check (see also answer to Anonymous Referee 1).

*Specific comments:*

*Line 11: why "even though"?*

We reformulated this entire sentence.

*Introduction: While you emphasize the importance of accounting for OH reactivity by its effects on the lifetime of VOCs oxidants in the atmosphere, there are additional central reasons why OH reactivity and BVOCs composition are important. Among these you can mention photochemical air-pollution and secondary aerosol formation.*

We agree with the referee that we arbitrarily limited ourselves to a subset of reasons about the important aspects of OH reactivity and BVOCs. This has been addressed in the revised manuscript with the suggestions of the referee: "Moreover, the oxidation of VOCs in the atmosphere can lead to the formation of secondary aerosol formation and may play a role in photochemical air pollution by affecting levels of oxidants and pollutants."

*Line 29-35: This paragraph use extensively "OH reactivity" which you define on line 111. It will be good to provide a brief explanation/definition before you discuss it here.*

This paragraph has been rewritten taking into account both referee's comments. The revised manuscript contains additional definition in this paragraph about OH reactivity

(inverse of OH lifetime, also named "total OH loss rate").

*Line 31: Can you provide reference/s to support this statement?*

We rephrased this sentence in the revised manuscript to reflect its intended original meaning. It is now: "To estimate the magnitude of missing OH chemical sinks, Kovacs and Brune (2001) started measuring total OH loss rates to compare with model results."

*Line 42: remove extra ")".*

Done.

*Line 50: "contradictory" – the use of this word seems inappropriate taking into account the rest of the paragraph.*

We replaced the word "contradictory" with the word "inconclusive".

*Line 57: What do you mean by "important" trees? Can you make this point clearer?*

We replaced "important" by "common" in the revised manuscript, as this is what was originally meant.

*Line 83: It's not a comparison between the "changes in biomass" and other uncertainties; you probably refer to the effect emanating from the former on OH reactivity. While this assumption seems reasonable can you indicate to what extent (e.g., in percentage) this effect could affect your results?*

Based on the work by Aalto et al. (2014), it was estimated that pine needles did not experience much growth during the measurement periods. It was also estimated that spruce needles were fully grown by 30 May and that the biomass difference is at most 20 % for the periods before (Branch S1). For birch, all the growth happen rapidly from no laves to the full measured dry weight of the biomass during the first period (Branch B1), so that the uncertainty due to the normalization of the biomass is very large at the beginning of that period and only the last five days the difference of the biomass is within 20 % of the measured dry weight. We therefore indicate this period with a

dashed line and added a note in the figure caption.

*Lines 141-142: Can you explain why you didn't use C3H8 for the calibration as described by Parplan et al. (2019) and Shina et al. (2008)?*

While a $C_3H_8$ calibration is described in Praplan et al. (2019), only the reactivity calibration for $\alpha$-pinene (as a proxy) was used, based of the input from a referee. In the present study, we also consider $\alpha$-pinene to have an average reactivity for the emissions, considering that some compounds will be more reactive (e.g. sesquiterpenes) and other compounds less reactive (e.g. alcohols). We estimated for the revised manuscript that the uncertainty of using $\alpha$-pinene as a proxy for the reactivity of the emissions is about 50 %. See also the answer to the comment from Anonymous Referee 1 about this.

*Line 147: Are the values in Eq. 3 referring to the calibration factors? Please indicate their meaning in the text.*

We included more information regarding the calibration factors in the text. We also streamlined this paragraph as Eq. 3 and 8 were redundant.

*Line 148: To what extent humidity can affect your results in addition to (/relative to) the dilution effect?*

We moved our description of the correction needed for RH difference between C2 and C3 measurements right after the sentence about dilution to address this referee's comment.

*Line 152: "Other correction factors need to be applied..." – Do you mean other than the presence of NOX and O3? If so, why did you write "However" in the next sentence?*

Our formulation was indeed misleading. In the revised manuscript we simply state why these corrections are not required.

*Lines 152-153: Is this because NOX and O3 are assumed to be effectively removed by*

*the HPZA-7000?*

Yes. This is now stated in the revised manuscript.

*Line 164: "OH levels" – do you mean the effect on RH levels?*

In the CRM instrument, RH levels in the reactor are directly associated with the production of OH in the reactor. This sentence has been reformulated to clarify is meaning to tell and the whole section about the Comparative Reactivity Method has been streamlined for further clarity.

*Line 168: "t o" should be "to".*

Fixed.

*Line 180: Eq. 8, please indicate clearly what is the meaning of the (calibration?) values.*

We have rewritten parts of this section and streamlined it in order to improve its clarity. It includes now a clearer explanation of how the calibration values have been derived.

*Lines 187-188: Please provide a reference to support this estimation.*

Using a similar setup, Owen et al. (1997) mention that the enclosure temperature can be used as a close estimate for the leaf temperature as it is 2 °C lower at most. We included this reference in the revised manuscript.

*Results and discussion: In this section you present 4 figures and 2 tables, but you only once refer to Fig. 5 and Table 1 and 2. Please try to refer the reader to each of these from the text.*

We have edited the results and discussion section(see also comments from Anonymous Referee 1). Proper references to figures and tables have been incorporated in the revised manuscript.

*Lines 209-2013: I suggest making this paragraph shorter and generally avoid summa-*

*rizing or discussing the results prior presenting the results themselves.*

This paragraph is indeed summarizing results before they are presented. We moved this paragraph towards the end of the section (see also answer to comment from Anonymous Referee 1).

*Table 1 caption: Acronyms - "Te" is not consistent with RH if the latter refers to measurements in the enclosure.*

We replaced "RH" with "RHe" as we always meant "RH in the enclosure". The table caption has been updated as well in the revised manuscript to clarify this.

*Line 214: Can you support it by statistical analysis?*

Our statement ("In general the missing OHRE fraction was higher in spring and decreased as the seasons proceeded.") is meant to describe qualitatively the observed trend of the missing OHRE fraction in Table 1. We are unsure what statistical analysis the referee wants to see.

*Line 235: GLVs was already defined before.*

We use only the abbreviation in this sentence in the revised manuscript.

*Line 236: "as well" appears twice.*

We have rewritten this unfortunately formulated sentence in the revised manuscript.

*Lines 245-247: Can you elaborate on why the fact that stress in your case was not driven by elevated temperature indicates that some of these (missing OHRE) are not terpenoids or oxidized volatile organic compounds?*

We have reformulated these sentences to clarify what it meant to tell: "In our study, these stress periods for pine, identified with GLVs emissions, are not related to elevated temperature (see section 3.5). Missing OHRE was generally higher during these periods, but as terpenoids were monitored, they cannot explain the stress-related emissions of reactivity. Some oxidised volatile organic compounds were also measured, but not methanol, formaldehyde, and acetaldehyde, for instance, which could contribute — at least in part — to the missing OHRE."

*Line 249: What do you mean by "follows qualitatively"? If you imply for correlation, can you calculate the Pearson correlation coefficient and present it to support this notion?*

Our statement meant to tell that even though the absolute values of TOHRE and COHRE differ, they follow a similar time evolution (e.g. maxima). This was an arbitrary statement based on visual examination, but thanks to the referee's input we now use the Pearson correlation coefficient in the revised manuscript to support our claim. While for spruce and pine a correlation could be established, it was not the case for birch data. The Pearson's correlation coefficient r is 0.88 and 0.78 pine and spruce, respectively (both with a p-value <0.01), for the periods when both COHRE and TOHRE are available. However, the Pearson's correlation coefficient r is much lower birch when both TOHRE and COHRE are available (0.02 with a p-value of 0.4). The text in the revised manuscript reflect these newly introduced statistics.

*Lines 253-255: What about aldehydes? How it compares with the findings by Hakola et al. (2017) about aldehyde emissions from Spruce?*

Hakola et al. (2017) found relatively high emissions of higher aldehydes, especially nonanal and decanal. In our study, these high emissions could not be observed, and their contribution to OHRE remained small.

*Lines 255-257: Does the higher emission of sesquiterpenes at the expense of GLVs under the different conditions can be supported by statistical analysis? Can you use regression to identify which of the parameters (temperature, radiation, precipitation) shows higher association with the emissions rate of the various VOCs?*

Sesquiterpenes are emitted from spruces in normal conditions while GLVs are induced by the stress. Sesquiterpene emission were not higher when GLVs were lower, but their

relative share on COHRE was higher. In our opinion, detailed studies on dependences of various VOCs on different parameters are out of scope of this manuscript and will be studied in separate manuscripts in future.

*Line 259: Why "However?*

We removed the transition word "However", when we edited this section for the revised manuscript.

*Lines 280-281: Can you indicate to what extent the constant blank subtraction could have affected your results?*

As we used a constant blank for the all the measurement periods, it cannot be excluded that the blank was underestimated or overestimated at times. Underestimation of the blank value affects in particular periods with low reactive emissions and lead to high relative missing reactivity values, which should be considered with caution as we tried to make clear in the manuscript.

*Line 282: "quantitatively" - Qualitatively? Can you calculate the Pearson correlation coefficient for all three species?*

Yes, "qualitatively" was meant here and we have now included Pearson correlation coefficients in the discussion for all three species (see also answer to comment above).

*Line 300: Please indicate the regression you have used to evaluate R. Did you try exponential regression (as is implied by Fig. 5)?*

We regret that we failed to explicitly mention in the text which regression we used to evaluate R. This has now been improved in the revised manuscript (see also answer to the comment from Anonymous Referee 1).

*Lines 301-303: Can you provide more information about the nature of the stress and the cause for the low TOHRE?*

Based on visual observations, we concluded that the nature of the stress was abiotic,

which we now mention in the revised manuscript. The low TOHRE values were due to low emissions in the cited periods, possibly due to cooler and cloudier weather.

*Line 305: Why not referring to Table 2?*

We refer to Table 2 (and Figure 5) in an earlier paragraph.

*Line 312: What do you mean by "good correlation with temperature"?*

We meant that in July the coefficient of correlation (R) with the temperature is 0.71. We have added this to the sentence in the revised manuscript.

*Line 316: "In a few cases was even slightly reduced." – this seems reasonable to me. I just want to make sure you have used exponential regression type for temperature.*

As mentioned previously, we used exponential regression and this is now explicitly mentioned in the revised text.

*Lines 321-323: A very general sentence - can you specify which factors ("other factors") and elaborate on that?*

We mainly meant abiotic stress factors as we demonstrate how they influence our results. This is now explicitly mentioned in the revised manuscript (see also answer to comment from Anonymous Referee 1).

*Line 332: "stress-induced" - Please specify stress type as much as possible.*

We rephrased this sentence to indicate that the stress was very likely drought.

*Lines 337-338: Can you provide an explanation for that?*

We mentioned in section 3.3 the study by Hakola et al. (2017), where similar observations were done regarding higher emissions of sesquiterpenes in the late summer. They speculated on the possible defensive role of sesquiterpenes, but the lack of any visible infestations of feeding herbivores indicated systemic defence mechanism rather than a direct one. We included this information in the discussion of the revised

manuscript, but we mention in the conclusions that this is observation is consistent with a previous study.

*Line 341: "highest" - Looks like a contradiction with the rest of the sentence*

The meaning of this sentence was indeed difficult to understand in the original formulation. What was meant is that if the CRM has a similar background for all species, normalizing with the biomass (smaller values for birch) will yield higher TOHRE values. The phrasing has been improved in the revised manuscript: "This is partly explained by total OH reactivity values measured close to the experimental background (independent of the tree species measured) and normalised by the smallest dry weight of the leaves or needles of all tree species."

*References*

Aalto, J., Kolari, P., Hari, P., Kerminen, V.-M., Schiestl-Aalto, P., Aaltonen, H., Levula, J., Siivola, E., Kulmala, M., and Bäck, J.: New foliage growth is a significant, unaccounted source for volatiles in boreal evergreen forests, Biogeosciences, 11, 1331–1344, 2014. doi:10.5194/bg-11-1331-2014.

*Hakola, H., Tarvainen, V., Praplan, A. P., Jaars, K., Hemmilä, M., Kulmala, M.,Bäck, J., and Hellén, H.: Terpenoid and carbonyl emissions from Norway sprucein Finland during the growing season, Atmos.Chem.Phys., 17, 3357–3370, 2017. doi:10.5194/acp-17-3357-2017.*

Owen, S., Boissard, C., Street, R. A., Duckham, S. C., Csiky, O., and Hewitt, C. N.: Screening of 18 Mediterranean plant species for volatile organic compound emissions, Atmos. Environ., 31, 101–117, 1997. doi:10.1016/S1352-2310(97)00078-2.

*Praplan, A. P., Tykkä, T., Chen, D., Boy, M., Taipale, D., Vakkari, V., Zhou, P., Petäjä, T.,and Hellén, H.: Long-term total OH reactivity measurements in a boreal forest, Atmos.Chem. Phys., 19, 14431–14453, 2019. doi:10.5194/acp-19-14431-2019.*

*Sinha, V., Williams, J., Crowley, J. N., and Lelieveld, J.: The Comparative Reactivity*

*Method – a new tool to measure total OH Reactivity in ambient air, Atmos. Chem.Phys., 8, 2213–2227, 2008. doi:10.5194/acp-8-2213-2008.*

---

## Author Response (AR2)

**Answers to Anonymous Referee #2's comments on "OH reactivity from different tree species: Investigating the missing reactivity in a boreal forest" by Praplan et al.**

Reviewer's comments are in italics and our answers are in normal font.

*I think that in general the authors have seriously addressed the comments on their manuscript. By doing so the revised version of the manuscript provides more complete information about the uncertainties which are associated with their methodologies and findings. In addition, the revised version presents the methodologies and discussions in a more complete and quantitative manner.*

We thank the referee for recognising the improvements made to the manuscript.

*I think that the authors should better address two specific points:*

*1. The removal of O3 and NOX in the dynamic branch enclosure - can you provide more details, either based on information provided by the manufacturer or based on the zero-air's mechanism of operation, that supports your assumption about NOX and O3 removal?*

The authors were unable to obtain specific information regarding the mechanism of operation from the manufacturer of the zero air generator. However, our assumption is based on our own observations. If $NO_x$ and/or $O_3$ would be present in the zero air, $C_3$ pyrrole levels in the CRM instrument would be lower than $C_2$ levels. This is what is observed for ambient measurements and without the corrections described in, for instance, Praplan et al. (2019), it would lead to negative values for total OH reactivity. This is because the air used for $C_2$ is going through platinum on aluminium pellets heated at roughly 450°C in addition, which removes VOCs, as well as $NO_x$, and $O_3$. In order to address the referee's comment, we added the following explanation to the manuscript: "Even though the specific mode of operation of the generator is not known, $C_2$ and $C_3$ levels are similar during periods of low emissions (e.g. at night) and during measurements of the blank chamber. If $NO_x$ and/or $O_3$ would be present in the generated zero air, $C_3$ would be lower than $C_2$, which is why these corrections are needed for ambient air measurements. However, this is not what is observed in the data presented here, which confirms that these corrections are not needed in this particular case."

*2. The fact that the missing OHRE fraction was higher in spring and decreased as the seasons proceeded appears to be one of the most important findings of this study (it is mentioned in both the Abstract and the overview of the Results and discussion). Considering the relatively high number of available observations (as indicated by Table 1) I would recommend to analyze this finding statistically. You could use a paired sample t-test to compare the means of the various distributions. You may also like to use K-S (Kolmogorov–Smirnov test) to use a nonparametric statistical test to compare the various distributions.*

According to the suggestion from the reviewer, we conducted statistical analysis of the missing OHRE fraction. We used paired sample t-test from the Python's package SciPy to compare the monthly means of missing OH reactivity reported in Table 1. We found that p-values were below 0.05 for month-to-month comparison for each tree species with the exception of the means for pine in June and July (p-value 0.52). We included a sentence in each sections from 3.2 to 3.4 to report these results.

**Complementary answer to a previous referee's comment**

In our previous answers to Anonymous Referee #1 regarding the comparison of our results for spruce with the ones from Nölscher at al. (2013), we wrote that due to the difference in the method used, the values reported for TOHRE had different units in both studies and we failed to compare both studies quantitatively.

However, shortly after we submitted our answers, it came to our attention that the volume of the chamber used in Nölscher et al. (2013) is reported in Bourtsoukidis et al. (2012). It is 15 L and by multiplying the TOHRE values in Nölscher et al. (2013) by the square of the chamber volume (in $m^3$), it is possible to convert the units of TOHRE values to $m^3$ $s^{-2}$ $g_{dw}^{-1}$. We allowed

ourselves to update the manuscript to include a quantitative comparison between both studies hoping that the editor sees this additional change favourably.

The following sentence is replacing the previous statement about the comparison in the latest version of the manuscript: "
[revised manuscript text omitted]
| hexanal | 1.5E-08 ($\pm$1.1E-08) | 1.5E-06 ($\pm$3.0E-06) | 2.8E-07 ($\pm$1.9E-07) | 8.4E-08 ($\pm$9.8E-08) | 1.2E-08 ($\pm$8.2E-09) | 5.0E-07 ($\pm$5.6E-07) |
| heptanal | 1.3E-08 ($\pm$1.3E-08) | 2.6E-08 ($\pm$3.2E-08) | n.d. | 2.8E-08 ($\pm$2.8E-08) | 2.1E-08 ($\pm$1.4E-08) | 2.0E-07 ($\pm$1.0E-07) |
| octanal | 2.3E-08 ($\pm$2.4E-08) | 3.7E-08 ($\pm$5.0E-08) | n.d. | 2.2E-08 ($\pm$2.9E-08) | 9.2E-09 ($\pm$7.3E-09) | 7.3E-08 ($\pm$4.2E-08) |
| nonanal | 4.2E-08 ($\pm$3.4E-08) | 9.3E-08 ($\pm$9.6E-08) | n.d. | 6.7E-08 ($\pm$6.7E-08) | 2.1E-08 ($\pm$1.6E-08) | 1.1E-07 ($\pm$6.0E-08) |
| decanal | 5.5E-08 ($\pm$3.2E-08) | 4.9E-08 ($\pm$5.5E-08) | n.d. | 2.3E-08 ($\pm$3.0E-08) | 1.4E-08 ($\pm$1.1E-08) | 4.9E-08 ($\pm$2.6E-08) |
| methacrolein | 2.2E-09 ($\pm$7.3E-09) | 1.5E-08 ($\pm$2.1E-08) | n.d. | 2.2E-08 ($\pm$2.1E-08) | 2.6E-09 ($\pm$3.1E-09) | 4.3E-08 ($\pm$2.7E-08) |
| 1-pentanol | 2.2E-10 ($\pm$3.9E-09) | 1.4E-09 ($\pm$2.3E-08) | n.d. | n.d. | 1.1E-10 ($\pm$1.9E-09) | 6.6E-09 ($\pm$4.3E-08) |
| 1-octen-3-ol | n.d. | n.d. | n.d. | n.d. | n.d. | n.d. |
| butyl acetate | n.d. | n.d. | n.d. | n.d. | n.d. | n.d. |
| bornyl acetate | 5.2E-08 ($\pm$1.5E-07) | 9.7E-08 ($\pm$2.2E-07) | 3.7E-09 ($\pm$2.1E-08) | 1.4E-08 ($\pm$1.5E-08) | 2.7E-08 ($\pm$1.4E-08) | 9.1E-08 ($\pm$1.2E-07) |
| propanoic acid | 6.5E-09 ($\pm$1.3E-08) | 2.3E-10 ($\pm$2.2E-09) | n.d. | n.d. | 7.1E-09 ($\pm$8.3E-09) | 6.7E-10 ($\pm$5.1E-09) |
| butanoic acid | 2.4E-08 ($\pm$1.6E-08) | 7.0E-09 ($\pm$8.6E-09) | n.d. | n.d. | 1.2E-08 ($\pm$7.4E-09) | 1.2E-08 ($\pm$1.5E-08) |
| isobutanoic acid | 2.2E-09 ($\pm$9.9E-09) | 1.9E-09 ($\pm$1.1E-08) | n.d. | n.d. | 1.1E-08 ($\pm$1.3E-08) | 1.0E-08 ($\pm$3.5E-08) |
| pentanoic acid | n.d. | n.d. | n.d. | n.d. | 1.4E-09 ($\pm$6.3E-09) | n.d. |
| isopentanoic acid | n.d. | n.d. | n.d. | n.d. | 3.7E-10 ($\pm$2.2E-09) | n.d. |
| hexanoic acid | n.d. | n.d. | n.d. | n.d. | 2.2E-10 ($\pm$3.8E-09) | n.d. |
| 4-methylpentanoic acid | n.d. | n.d. | n.d. | n.d. | n.d. | n.d. |
| heptanoicacid | n.d. | n.d. | n.d. | n.d. | n.d. | n.d. |

[a] quantified as $\Delta^3$-carene  [b] quantified as bornylacetate  [c] quantified as terpinolene  [d] quantified as isolongifolene  [e] quantified as $\beta$-farnesene  [f] quantified as $\beta$-caryophyllene  [g] quantified as $\beta$-caryophyllene or isolongifolene  [h] quantified as longicyclene

Table D2: Averages of individual compounds' OH reactivity of the emissions, OHRE [$m^3\,s^{-2}\,g_{dw}^{-1}$], with standard deviations (in brackets) for the different measurement periods for spruce; 'n.d.' means 'not detected' and 'n.m.' means 'not measured'

| | Period S1.a 11–14 May | Period S1.b 19–24 May | Period S2 5–13 July | Period S3.a 16–23 August | Period S3.b 6–13 September |
|---|---|---|---|---|---|
| isoprene | 1.3E-07 ($\pm$1.9E-07) | 1.0E-06 ($\pm$2.3E-06) | 7.3E-07 ($\pm$1.0E-06) | 8.9E-08 ($\pm$1.2E-07) | 6.4E-08 ($\pm$7.2E-08) |
| MBO | 1.8E-07 ($\pm$2.8E-07) | 6.5E-07 ($\pm$1.2E-06) | 8.3E-07 ($\pm$1.3E-06) | 2.1E-07 ($\pm$2.8E-07) | 1.1E-07 ($\pm$1.3E-07) |
| $\alpha$-pinene | 6.9E-06 ($\pm$1.1E-05) | 9.7E-07 ($\pm$2.0E-06) | 1.8E-05 ($\pm$3.8E-05) | 3.8E-07 ($\pm$3.9E-07) | 1.1E-06 ($\pm$9.7E-07) |
| $\beta$-pinene | 1.4E-05 ($\pm$2.2E-05) | 8.9E-07 ($\pm$2.5E-06) | 1.4E-05 ($\pm$1.9E-05) | 2.5E-07 ($\pm$2.2E-07) | 1.3E-06 ($\pm$1.2E-06) |
| camphene | 3.7E-06 ($\pm$6.0E-06) | 3.4E-07 ($\pm$6.5E-07) | 2.7E-05 ($\pm$6.0E-05) | 4.8E-07 ($\pm$7.7E-07) | 1.2E-06 ($\pm$9.8E-07) |
| $\Delta^3$-carene | 3.8E-06 ($\pm$8.6E-06) | 1.7E-07 ($\pm$4.4E-07) | 5.9E-06 ($\pm$7.3E-06) | 6.9E-08 ($\pm$3.9E-08) | 2.2E-07 ($\pm$2.2E-07) |
| $\beta$-phellandrene[a] | 1.5E-05 ($\pm$2.3E-05) | 7.0E-07 ($\pm$1.9E-06) | 8.0E-06 ($\pm$1.4E-05) | 9.2E-08 ($\pm$5.5E-08) | 9.8E-07 ($\pm$1.1E-06) |
| *p*-cymene | 2.2E-07 ($\pm$4.7E-07) | 1.5E-08 ($\pm$3.1E-08) | 3.8E-07 ($\pm$7.7E-07) | 3.8E-09 ($\pm$2.5E-09) | 7.2E-09 ($\pm$5.7E-09) |
| 1,8-cineol | 6.0E-07 ($\pm$7.2E-07) | 6.0E-08 ($\pm$1.1E-07) | 4.9E-06 ($\pm$1.2E-05) | 3.4E-08 ($\pm$2.6E-08) | 2.0E-07 ($\pm$1.8E-07) |
| limonene | 3.4E-05 ($\pm$4.4E-05) | 3.6E-06 ($\pm$6.2E-06) | 1.1E-04 ($\pm$2.5E-04) | 2.9E-06 ($\pm$2.4E-06) | 1.7E-05 ($\pm$1.3E-05) |
| terpinolene | 4.1E-06 ($\pm$8.1E-06) | 2.7E-07 ($\pm$5.8E-07) | 6.2E-06 ($\pm$1.8E-05) | 6.1E-08 ($\pm$4.2E-08) | 1.8E-07 ($\pm$2.
[revised manuscript text omitted]

585